# Protein absorption in the zebrafish gut is regulated by interactions between lysosome rich enterocytes and the microbiome

Laura Childers[1], Jieun Park[2,3], Siyao Wang[1], Richard Liu[1], Robert Barry[4], Stephen A Watts[4], John F Rawls[5]*, Michel Bagnat[1]*

[1]Department of Cell Biology, Duke University, Durham, Durham, United States; [2]Neuroscience Center, University of North Carolina, Chapel Hill, United States; [3]Carolina Institute of Developmental Disabilities, Chapel Hill, United States; [4]Department of Biology, University of Alabama at Birmingham, Birmingham, United States; [5]Department of Molecular Genetics and Genomics, Duke University, Durham, United States

## eLife Assessment

In this study, the authors use the zebrafish to investigate how the microbiome affects a specialized gut cell called the lysosome rich enterocyte. They use a combination of functional assays for protein absorption, gnotobiotic manipulations and single-cell RNA-seq. The findings in the paper are considered **important** and the results are **convincing**.

*For correspondence:
john.rawls@duke.edu (JFR);
michel.bagnat@duke.edu (MB)

**Abstract** Dietary protein absorption in neonatal mammals and fishes relies on the function of a specialized and conserved population of highly absorptive lysosome-rich enterocytes (LREs). The gut microbiome has been shown to enhance absorption of nutrients, such as lipids, by intestinal epithelial cells. However, whether protein absorption is also affected by the gut microbiome is poorly understood. Here, we investigate connections between protein absorption and microbes in the zebrafish gut. Using live microscopy-based quantitative assays, we find that microbes slow the pace of protein uptake and degradation in LREs. While microbes do not affect the number of absorbing LRE cells, microbes lower the expression of endocytic and protein digestion machinery in LREs. Using transgene-assisted cell isolation and single cell RNA-sequencing, we characterize all intestinal cells that take up dietary protein. We find that microbes affect expression of bacteria-sensing and metabolic pathways in LREs, and that some secretory cell types also take up protein and share components of protein uptake and digestion machinery with LREs. Using custom-formulated diets, we investigated the influence of diet and LRE activity on the gut microbiome. Impaired protein uptake activity in LREs, along with a protein-deficient diet, alters the microbial community and leads to an increased abundance of bacterial genera that have the capacity to reduce protein uptake in LREs. Together, these results reveal that diet-dependent reciprocal interactions between LREs and the gut microbiome regulate protein absorption.

## Introduction

The ability of the intestine to efficiently absorb nutrients from the diet is influenced significantly by the microbiome that it harbors (*Wilson et al., 2020*; *Kau et al., 2011*). Most dietary nutrients, including proteins, lipids, and carbohydrates, are absorbed by the small intestinal epithelium after luminal digestion. In zebrafish and mice, microbiome colonization enhances small intestinal absorption of dietary lipids compared to germ-free (GF) animals (*Semova et al., 2012*; *Martinez-Guryn et al., 2018*). Gut microbiomes are also known to increase dietary energy harvest by fermenting complex carbohydrates into short-chain fatty acids that can be absorbed by the gut epithelium (*Cholan et al., 2020*). There is evidence that *Drosophila*-associated microbes absorb dietary proteins and amino acids and are, in turn, ingested and metabolized by the host, promoting survival under dietary protein-limited conditions (*Keebaugh et al., 2018*; *Yamada et al., 2015*; *Lesperance and Broderick, 2020*). Furthermore, amino acids secreted by the *Drosophila* microbiome have been shown to modify feeding behavior (*Kim et al., 2021*). However, whether the microbiome affects intestinal absorption of dietary proteins remains poorly understood. The importance of this question is underscored by the gut microbiome's links to protein malnutrition diseases. Children with kwashiorkor, a disease caused by severe protein malnutrition, have significantly altered gut microbiomes that promote weight loss when transplanted into GF mice (*Smith et al., 2013*). These studies suggest that protein deprivation may cause a gut microbial community to develop that further exacerbates the effects of the disease. Defining the reciprocal interactions between intestinal physiology, microbiome, and dietary protein nutrition is, therefore, an important research goal.

In neonatal mammals and fishes, dietary protein absorption is dependent on the function of a specialized population of epithelial cells in the ileal region of the small intestine originally described as vacuolated or neonatal enterocytes (*Kraehenbuhl and Campiche, 1969*; *Rodríguez-Fraticelli et al., 2015*; *Wallace et al., 2005*; *Gonnella and Neutra, 1984*; *Wilson et al., 1987*; *Rombout et al., 1985*; *Graney, 1968*). Recent work in zebrafish and mice showed that these cells, which we refer to as LREs, are highly endocytic and specialize in the uptake of luminal proteins that they then digest in giant lysosomal vacuoles (*Park et al., 2019*). In mammals, LREs are present only during suckling stages and are lost at weaning (*Harper et al., 2011*), whereas in fishes they are retained through adult life (*Park et al., 2019*; *Noaillac-Depeyre and Gas, 1976*; *Stroband and Debets, 1978*). In zebrafish, digestive processes can be observed live in the transparent larvae following gavage with fluorescent cargoes directly into the intestinal lumen (*Park et al., 2019*; *Rodríguez-Fraticelli et al., 2015*). Using this assay, LREs were shown to internalize proteins but not lipids from the intestinal lumen (*Park et al., 2019*).

LREs internalize luminal proteins using an endocytic complex composed of the scavenger receptor cubilin (Cubn), the transmembrane linker amnionless (Amn), and endocytic clathrin adaptor Dab2 (*Park et al., 2019*). Loss of these components severely reduces their capacity to take up luminal proteins, leading to stunted growth, poor survival, and intestinal edema reminiscent of kwashiorkor (*Park et al., 2019*). LREs are broadly present in vertebrates and likely also in lower chordates (*Nakayama et al., 2019*; *Yonge, 1923*), suggesting LRE development and physiology are ancient and important aspects of intestinal function. However, whether gut microbes play a role in LRE-dependent processes or the pathobiology of LRE deficiency is not known.

Previous work suggested that microbes affect uptake and degradation kinetics in LREs. Specifically, zebrafish larvae colonized with a complex gut microbiome (ex-GF conventionalized or CV) had electron-dense material in the lysosomes of LREs that was not detected in GF larvae (*Rawls et al., 2004*). Consistent with these findings, CV larvae immersed in horse radish peroxidase (HRP) had increased HRP accumulation in LREs (*Bates et al., 2006*). However, it was unclear from those studies if microbes caused the material to accumulate in LREs by increasing the rate of LRE uptake, decreasing LRE degradation, or both.

Here, we investigate how interactions between LREs, the gut microbiome, and diet affect host nutrition in zebrafish larvae. We demonstrate that the gut microbiome reduces the rates of protein uptake and degradation in LREs. We present single-cell RNA sequencing (scRNA-seq) data that represents all major intestinal cell types, uncovering cell populations capable of protein uptake, and the effects of microbes across the gut. Using monoassociation experiments, we dissected the effects of specific microbial strains on LREs and found that *Vibrio cholerae* colonization strongly reduces LRE activity. Finally, using 16 S rRNA gene sequencing and custom-formulated diets, we found that

dietary protein content and LRE activity also affect the gut microbiome composition. Together, our results uncover significant interactions between LREs, diet, and gut microbes that regulates host nutrition.

## Results

### Gut microbiome slows uptake and degradation kinetics in LREs

We first investigated if microbes affect protein absorption in LREs. To do this, we used established methods (*Pham et al., 2008*) to rear zebrafish to the larval stage in gnotobiotic conditions to compare LRE activity in GF and ex-GF conventionalized (CV) conditions. In these experiments, a portion of the GF cohort was conventionalized with microbes at 3 d post fertilization (dpf), while the rest remained in the GF condition until the experimental endpoint of 6 dpf. At that point, we gavaged GF and CV larvae with fluorescent soluble cargoes and imaged uptake in the LRE region using confocal microscopy (*Figure 1A*; *Park et al., 2019*; *Shaner et al., 2004*; *Cocchiaro and Rawls, 2013*). To test how the microbiome affects protein uptake in LREs, we gavaged larvae with the purified fluorescent protein mCherry (*Shaner et al., 2004*). Previous work showed that LREs readily take up mCherry following gavage with fluorescence peaking in the anterior LREs (*Park et al., 2019*). We found that GF and CV larvae rapidly took up mCherry in the LREs, absorbing detectable levels of mCherry within 5 min of gavage (*Figure 1B-C*). LREs progressively became more saturated with mCherry between 5–60 min post-gavage in both conditions (*Figure 1B-C*). However, mCherry fluorescence peaked in the anterior LREs of GF larvae by 40 min PG (*Figure 1—figure supplement 1*), but the peak did not emerge until 60 min PG in CV larvae (*Figure 1—figure supplement 1*). Between 5 and 60 min PG, the anterior LREs of GF larvae accumulated mCherry at a significantly faster rate than those in CV larvae (*Figure 1—figure supplement 1*). Across the entire LRE region, GF larvae took up significantly more mCherry than CV larvae by 1 hr PG (*Figure 1D*). However, LREs in CV larvae eventually reached a similar level of mCherry uptake by 5 hr PG (*Figure 1E*). Over the 1–5 hr time course, the anterior LREs in CV larvae gradually increased in mCherry saturation, while mCherry saturation remained at a stable, high level in GF larvae (*Figure 1—figure supplement 1*). We tested if the gut microbiome reduces mCherry uptake in LREs by lowering its concentration in the lumen. That did not appear to be the case because luminal mCherry concentrations were equivalent in GF and CV larvae at 1 and 5 hr PG (*Figure 1—figure supplement 1*), and the microbiome did not degrade mCherry over time (*Figure 1—figure supplement 1*). We observed a similar trend using Lucifer Yellow (LY) as a fluid phase endocytic tracer (*Swanson et al., 1985*; *Park et al., 2019*). GF larvae took up significantly more LY than CV larvae by 1 hr PG (*Figure 1F*) but accumulated similar amounts by 3 hr PG (*Figure 1—figure supplement 1*). These results suggest that microbial colonization reduces the rate of endocytosis in the LRE region.

We hypothesized that microbial burden may influence the rate of luminal uptake and that a threshold density may be needed for microbes to affect protein-uptake kinetics in LREs. To investigate this possibility, we compared mCherry uptake activity between CV larvae with different microbial densities. At 1 hr PG, mCherry uptake was not reduced in CV larvae when the density was $3\times10^5$ CFU/mL in the gnotobiotic zebrafish media (GZM) (*Figure 1—figure supplement 1*). However, a higher density of $3\times10^6$ CFU/mL in the GZM was sufficient to significantly reduce mCherry uptake in CV larvae compared to GF larvae (*Figure 1—figure supplement 1*).

LREs degrade much of the protein they take up from the intestinal lumen within their lysosomal vacuole (*Park et al., 2019*). To test if microbes affect the protein degradation process, we employed a pulse-chase assay (*Park et al., 2019*) to compare the rate of protein degradation between GF and CV larvae (*Figure 2A*). At 6 dpf, larvae were gavaged with mTurquoise (25 mg/mL), a pH-insensitive protein that degrades rapidly in LREs (*Figure 2—figure supplement 1*). One hour after gavage, the remaining luminal mTurquoise was flushed with PBS, and live larvae were imaged with confocal microscopy over the course of 1 hr. In both GF and CV larvae, mTurquoise degraded over the 1 hr imaging time course (*Figure 2B-C*), with the degradation process occurring at a significantly faster rate in GF compared to CV LREs (*Figure 2D*). Together, these uptake and degradation kinetics assays demonstrate that the microbiome reduces the rate of cargo uptake and lysosomal protein degradation in LREs.

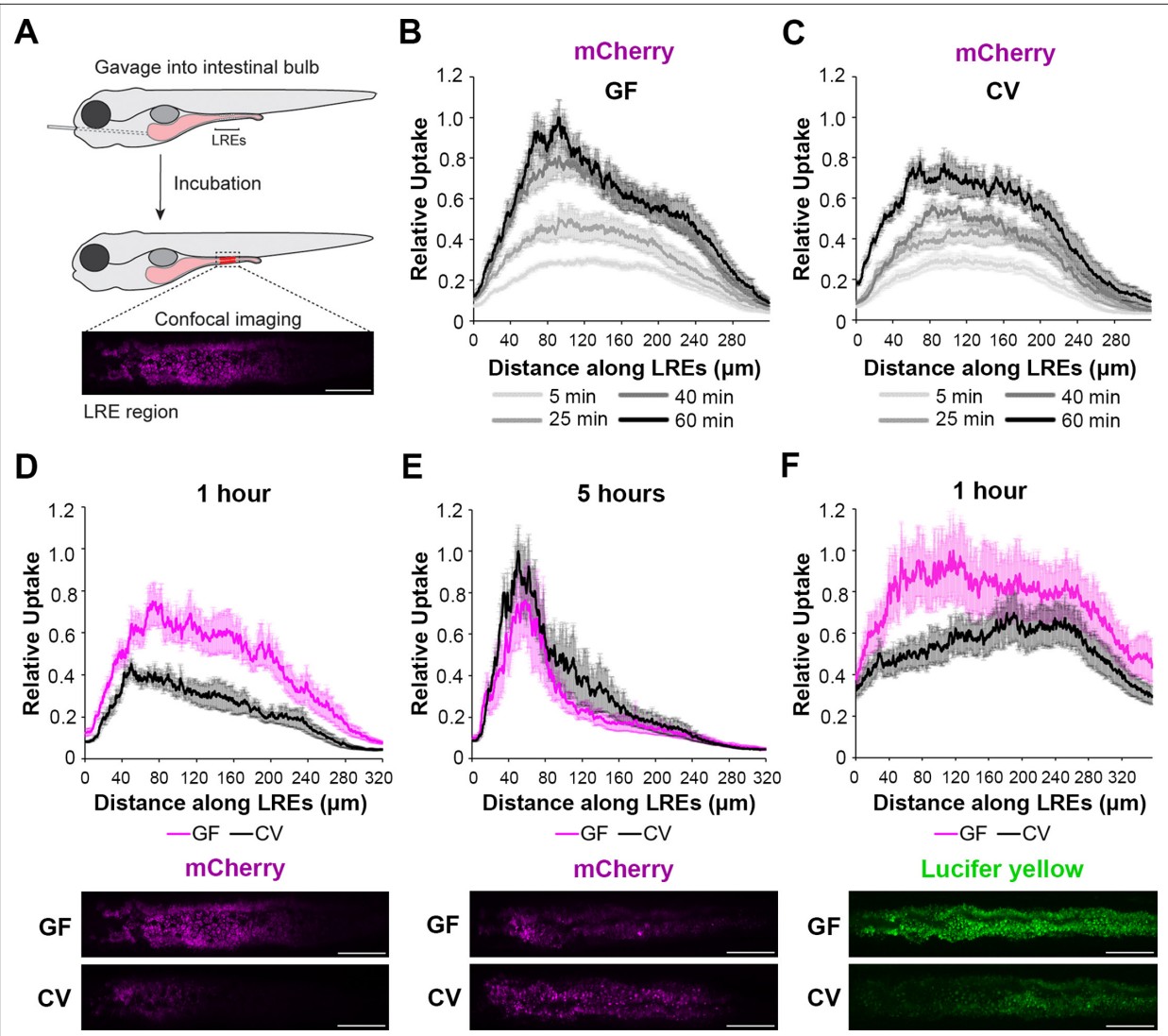

**Figure 1.** Microbes slow the rate of soluble cargo uptake in lysosome-rich enterocytes (LREs). (**A**) Cartoon depicting experimental design of the gavage assay in GF and CV larvae. Following derivation under conventional (CV) or germ-free (GF) conditions, 6 dpf larvae were gavaged, and uptake of luminal cargoes by LREs was measured by confocal microscopy in the LRE region (approximately 300 μm in length). (**B, C**) Plots of normalized mCherry fluorescence intensity along the LRE region over time in 6 dpf GF (**B**) and CV (**C**) larvae. Minutes after gavage, LREs rapidly took up and quickly accumulate mCherry in GF larvae. The anterior LREs approached full saturation by 40 min post gavage. Cargo uptake was slower in CV larvae, and anterior LREs did not reach saturation by 40 min post gavage. (**D, E**) Top: Plots of normalized mCherry fluorescence intensity along the LRE region of GF and CV larvae at 1 (**D**) and 5 (**E**) hr PG. GF larvae internalized significantly more mCherry than CV larvae (2-way ANOVA, p<0.0001, n=8–10) 1 hr PG, and CV larvae reached a similar level of mCherry accumulation to GF larvae by 5 hr PG (two-way ANOVA, p=0.137, n=8–11). Bottom: Representative confocal images showing mCherry signal in the LRE region (scale bars = 50 μm). (**F**) Top: Plot of normalized lucifer yellow fluorescence intensity along the LRE region of GF and CV larvae at 1 hr post gavage. LREs in GF larvae internalized significantly more Lucifer yellow than CV larvae by 1 hr post gavage (two-way ANOVA, p<0.0001, n=8).

The online version of this article includes the following figure supplement(s) for figure 1:

**Figure supplement 1.** Regional differences and impacts of microbial density on cargo uptake by lysosome-rich enterocytes (LREs).

## Identification of two microbe-independent LRE cell clusters and a microbe-dependent cloaca cluster

Given the differences we observed in LRE kinetics between GF and CV larvae, we wanted to investigate how the gut microbiome impacts gene expression programs in LREs and other intestinal cells. To do so, we prepared GF or CV *TgBAC(cldn15la-GFP)* transgenic larvae (*Alvers et al., 2014*), in which all intestinal cells are GFP-positive, and then gavaged them with mCherry to label LREs and other cells

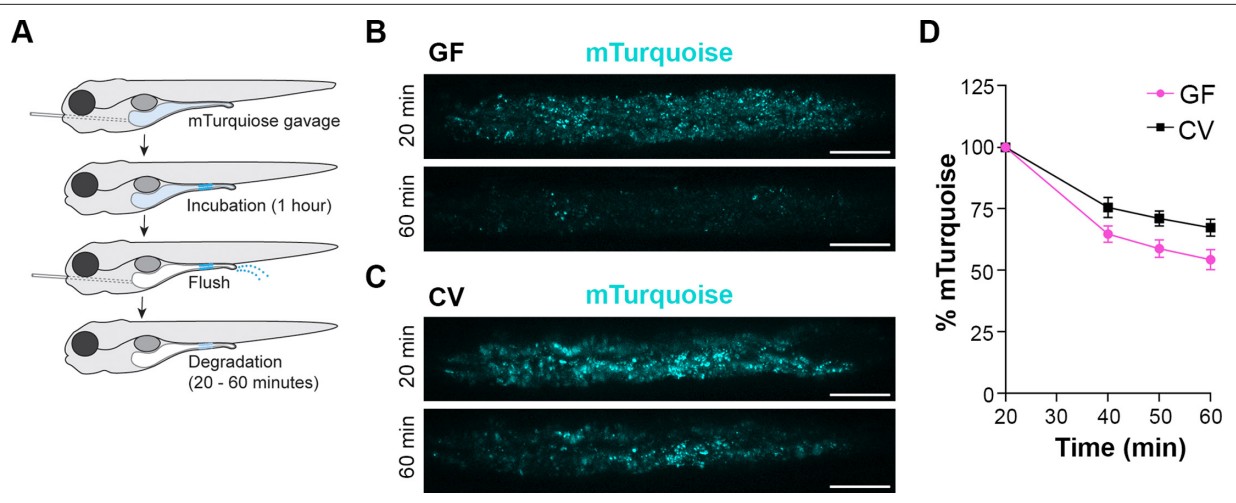

**Figure 2.** Lysosome-rich enterocyte (LRE) protein degradation activity is reduced by microbes. (**A**) Cartoon depicting the experimental design of pulse-chase protein uptake and degradation assay. At 6 dpf, germ-free (GF) and conventional (CV) larvae were gavaged with mTurquoise (25 mg/mL), incubated for 1 hr, and then flushed with PBS to remove luminal mTurquoise. LRE degradation of mTurquoise was measured by confocal microscopy over time. (**B, C**) Confocal images of mTurquoise fluorescence in the LRE region after flushing in GF (**B**) and CV (**C**) larvae (scale bars = 50 µm). (**D**) Plot showing the degradation of mTurquoise fluorescence (%) in the LRE region over time. Degradation occurred at a significantly faster rate in GF than CV larvae from 20 to 60 min post gavage (Simple linear regression, p=0.0167, n=6).

The online version of this article includes the following figure supplement(s) for figure 2:

**Figure supplement 1.** mTurquoise degradation kinetics in lysosome-rich enterocytes (LREs).

at 6 dpf (*Figure 3A*). At 3 hr PG, we isolated mCherry-positive/GFP-positive and mCherry-negative/GFP-positive cells by fluorescent-activated cell sorting (FACS) as previously described (*Park et al., 2019*) and processed them for scRNA-seq.

Clustering analysis with Seurat (*Satija et al., 2015*) revealed seventeen cell clusters, including two LRE clusters (*Figure 3B*). Each of these clusters was characterized by transcriptomic signatures that distinguished them from other clusters (*Figure 3C*) (*Supplementary file 1*; *Supplementary file 2*; *Supplementary file 3*). Ileocytes are an ileal enterocyte population that specialize in bile salt recycling (*Wen et al., 2021*). Interestingly, *fatty acid binding protein 6* (*fabp6*), a top ileocyte cluster marker, was also expressed in both LRE clusters (*Figure 3C*). Using transgenic reporters, previous studies showed that *fabp6* expression is highest in the ileocytes, but expression was also detected in the anterior LRE region (*Wen et al., 2021*; *Lickwar et al., 2017*). We, therefore, termed the LRE cluster with the higher *fabp6* expression in our scRNAseq dataset 'anterior LREs' and the LRE cluster with lower *fabp6* expression 'posterior LREs.' In total, this dataset includes 131 GF and 199 CV anterior LREs, along with 359 GF and 381 CV posterior LREs (*Figure 3—figure supplement 1*).

GF and CV larvae had proportionally similar numbers of cells in each cluster, including LREs (*Figure 3—figure supplement 1*), with the exception of a cluster we called Cloaca 3, which was only present in the CV condition (*Figure 3D*). GO term analysis revealed that Cloaca 3 consisted of IECs that are enriched in pathways related to host defense, response to bacteria, and iron ion transport (*Figure 3—figure supplement 1*). Several of the genes involved in bacterial response were top cluster markers for Cloaca 3 (*Figure 3—figure supplement 1*). These include genes involved in sensing bacteria through lipopolysaccharide-binding, neutrophil recruitment, and inflammation (*Kanther et al., 2011*). Their expression was markedly higher in Cloaca 3 than in neighboring clusters (*Figure 3—figure supplement 1*). For example, one of the Cloaca 3 marker genes was serum amyloid a (*saa*), which is known to be induced by microbiota in the distal intestine and cloaca (*Murdoch et al., 2019*). These results suggest that the development of Cloaca 3 cells is stimulated by the microbiome and functions in microbial sensing and immune response.

Our scRNAseq data indicated that LRE numbers were similar in the presence or absence of the gut microbiome (*Figure 3—figure supplement 1*). To quantitatively determine if microbes affect the number of LREs, we gavaged DQ-red BSA to mark the LRE lysosomes (*Marwaha and Sharma, 2017*) in GF and CV larvae and computationally segmented LREs using ilastik (*Sommer et al., 2011*;

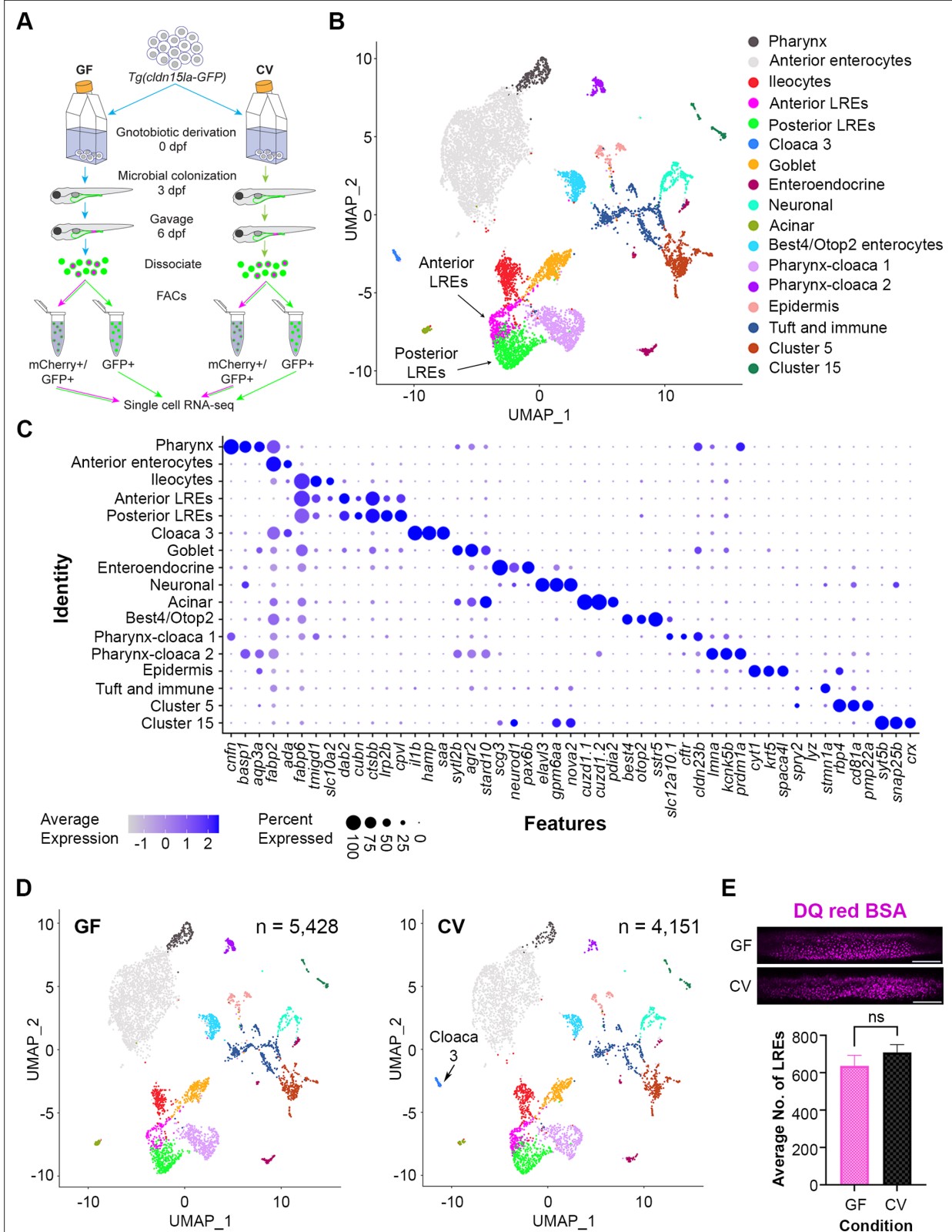

**Figure 3.** Single-cell clustering reveals anterior and posterior lysosome-rich enterocytes (LREs), microbially-responsive cloaca cells. (**A**) Cartoon depicting experimental design for transcriptomic profiling of intestinal cells in germ-free (GF) and conventional (CV) larvae. GF and CV larvae expressing *Tg(cldn15la-GFP)* to label all intestinal epithelial cells (IECs) were raised to 6 dpf in gnotobiotic conditions and then gavaged with mCherry (1.25 mg/mL). Cells from dissociated larvae were fluorescent-activated cell sorting (FACs) sorted to isolate GFP-positive/mCherry-positive from GFP-positive/

*Figure 3 continued on next page*

*Figure 3 continued*

mCherry-negative populations prior to single cell sequencing. (**B**) UMAP projection of cells color-coded by cluster identity. (**C**) Dot plot of top cluster markers in each cluster. Average expression of the marker gene in each cell cluster is signified by the color gradient. Dot size indicates the percentage of cells in each cluster expressing the marker. (**D**) UMAP projection of cells color-coded by cluster identity in the GF (left) and CV (right) datasets. The Cloaca 3 cluster only appeared in the CV dataset. (**E**) Bar plot showing that the average number of LREs in the GF and CV larvae was not significantly different (two-tailed t-test, p=0.33, n=10). LREs were labeled by gavaging with DQ red BSA (50 μg/mL) in a separate experiment, then quantified. Images show DQ red BSA marking LRE lysosomal vacuoles (scale bar = 50 μm).

The online version of this article includes the following figure supplement(s) for figure 3:

**Figure supplement 1.** Cell counts and features of conventional (CV)-specific Cloaca 3 cells.

*Figure 3—figure supplement 1*). This experiment revealed that there is not a significant difference in the number of active LREs between GF and CV larvae (*Figure 3E*). Together, these data show that the development of most intestinal cell clusters, including LREs, is not dependent on the gut microbiome. The notable exception is Cloaca 3, which only appeared in the presence of gut microbes.

## Identification of cell types with protein uptake capacity

Next, we explored the effects of the gut microbiome on mCherry uptake throughout the gut. Surprisingly, scRNA-seq analysis revealed that mCherry is internalized by several cell types in addition to LREs (*Figure 4A–B*). While LREs had the highest percentage of mCherry-positive cells, other mCherry-positive cell types included ileocytes, goblet, acinar, enteroendocrine, immune, and *best4/otop2* cells in both GF and CV conditions (*Figure 4C*). Notably, anterior enterocytes, pharynx, and cloaca clusters contained extremely low levels of mCherry-positive cells (*Figure 4C*), showing that these cell types have very low protein uptake activity.

The gut microbiome increased the proportions of mCherry-positive cells in several clusters (*Figure 4D*). This effect could be observed in secretory cells such as enteroendocrine, goblet cells, and acinar cells, which increased by 25%, 11%, and 8%, respectively. The gut microbiome also increased the proportion of mCherry-positive neurons by 13%. Notably, the proportion of mCherry-positive anterior (84–91%) and posterior (98–97%) LREs was extremely high in both GF and CV conditions, respectively (*Figure 4C*). However, the gut microbiome had mixed effects on the number of mCherry-positive LREs. While anterior LREs had a slightly higher proportion of mCherry-positive cells in the CV condition (7% increase), microbes caused a small decrease in the proportion of mCherry-positive posterior LREs (1% decrease) (*Figure 4C–D*). Regardless of microbial colonization, the proportion of mCherry-positive LREs remained high, underscoring the robust protein-uptake program in these cells.

Since the microbiome increased the proportion of mCherry-positive cells in many clusters, we proceeded to investigate the transcriptional program they hold in common. There were many differentially expressed genes between aggregated mCherry-positive and mCherry-negative cells in the CV condition (*Figure 4E*). The top upregulated genes in aggregated mCherry-positive cells (*ctsbb*, *lrp2b*, *ctsl.1*, *dab2*, *cpvl*, *cubn*, *tdo2b*, *lgmn*, *fabp6*, *sptbn5*) were also significant LRE cluster markers (*Figure 4—figure supplement 1*). On the other hand, the most upregulated genes in aggregated mCherry-negative cells (*fabp2*, *chia.2*, *fabp1b.1*, *apobb.1*, *apoa4b.1*, *apoa1a*, *tm4sf4*, *afp4.1*, *apoc2*, *apoc1*) were also significant anterior enterocyte cluster markers (*Figure 4—figure supplement 1*). This presumably reflects the high relative abundance of LRE and anterior enterocytes in the mCherry-positive and mCherry-negative groups, respectively (*Figure 4C*). However, expression of the top differentially expressed genes also clearly delineated mCherry-positive and mCherry-negative cells in secretory cell clusters, including goblet, EEC, and acinar cells (*Figure 4F*, *Figure 4—figure supplement 1*). Differential expression of these genes between mCherry-positive and negative cells was significant in these clusters (*Supplementary file 4*). Upregulated genes in mCherry-negative cells were highly enriched in anterior enterocytes and mCherry-negative cloaca 3, acinar, pharynx, and *best4/otop2* cells (*Figure 4—figure supplement 1*). These data suggest that intact protein uptake in non-LRE cells is associated with the expression of a subset of markers involved in protein uptake in LREs.

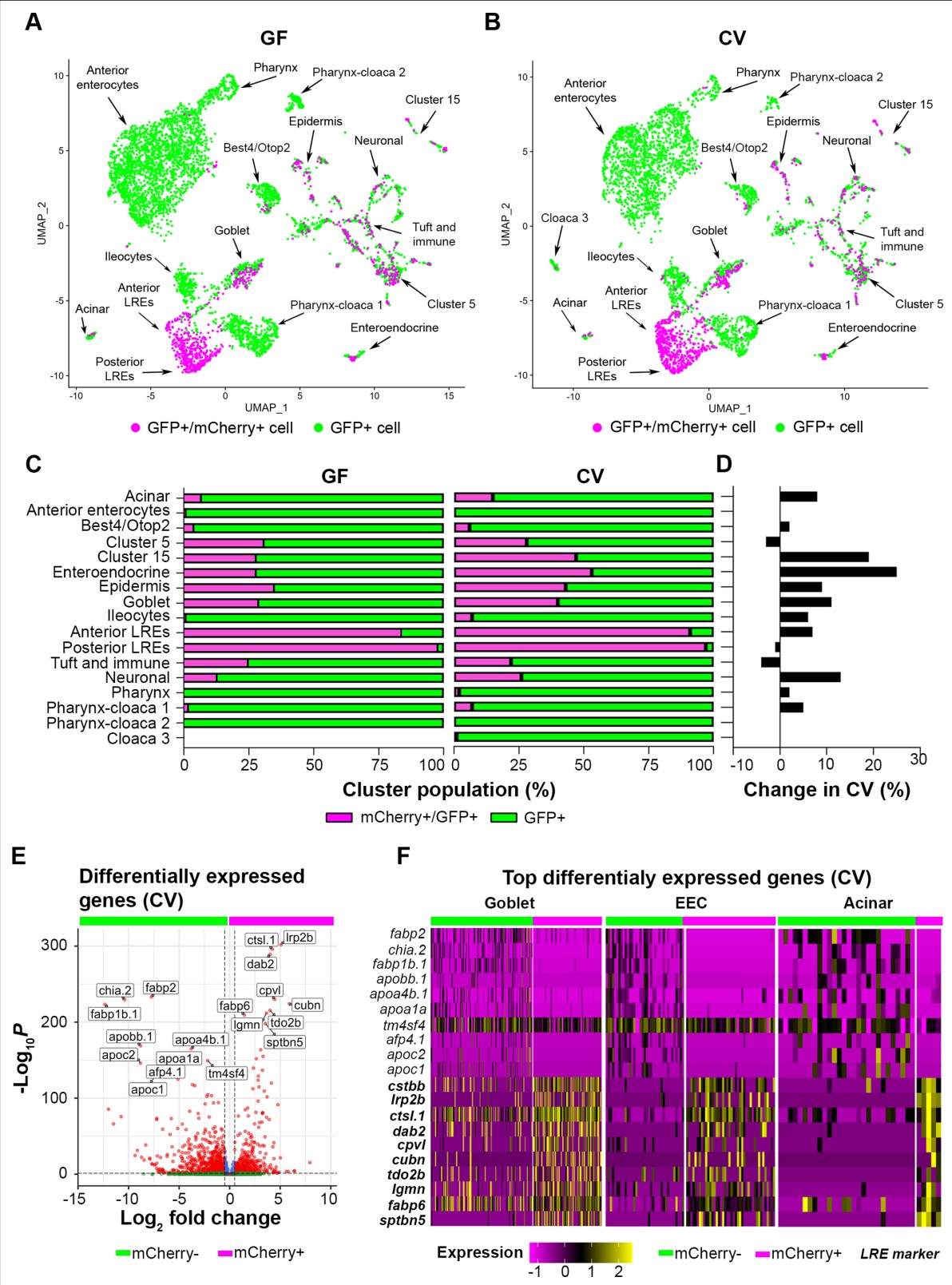

**Figure 4.** Uptake of mCherry occurs in cells enriched in lysosome-rich enterocyte (LRE) markers. (**A, B**) UMAP projections highlighting mCherry-positive/GFP-positive cells (magenta) and GFP-positive cells (green) in the germ-free (GF) and conventional (CV) datasets. (**C**) Bar plots portray the percentage of mCherry-positive and mCherry-negative cells in the GF and CV datasets. Bar color indicates the proportion of mCherry-positive (magenta) and mCherry-negative (green) cells in each cluster. (**D**) Bar plot displays the difference in the proportion of mCherry-positive cells in the CV compared to the GF

*Figure 4 continued on next page*

*Figure 4 continued*

dataset. Positive values show that the proportion of mCherry-positive cells were higher in the CV dataset. (**E**) Volcano plot shows differentially expressed genes between mCherry-positive and mCherry-negative cells in the CV dataset. The x-axis displays the log fold change in expression between mCherry-positive and mCherry-negative cells, with positive values showing enhanced expression in mCherry-positive cells and negative values showing higher expression in mCherry-negative cells. Red points are genes with significantly different expression (padj <0.05) and high fold change (log2FC < - 0.05, log2FC >0.05). (**F**) Heatmap displays the expression of the top markers for mCherry-positive and mCherry-negative cells in goblet, EEC, and acinar clusters. The color bar at the top indicates mCherry-positive (magenta) and mCherry-negative (green) cell types. Expression level is highlighted with a color gradient. mCherry-positive cells showed higher expression of *dab2* and other LRE-enriched endocytic markers (bolded), whereas mCherry-negative cells express typical anterior enterocyte markers such as *fabp2*.

The online version of this article includes the following figure supplement(s) for figure 4:

**Figure supplement 1.** Lysosome-rich enterocyte (LRE) and anterior enterocyte marker expression delineates mCherry-positive and mCherry-negative cells.

## Regional expression patterns and responses to the microbiome in LRE clusters

Despite similarities between LREs and mCherry-positive cells from other clusters, LREs maintained distinct transcriptional patterning that distinguished them from other cell types, including close clusters (*Figure 5A*). For example, KEGG pathway analysis confirmed that the lysosome pathway was strongly upregulated in anterior and posterior LREs in both GF and CV conditions (*Figure 5—figure supplement 1*), distinguishing them from their closest cluster neighbors (*Figure 5B*, *Figure 5—figure supplement 1*). In addition, both anterior and posterior LREs strongly expressed their characteristic endocytic machinery, composed of *cubn*, *dab2*, and *amn* (*Figure 5—figure supplement 1*).

Anterior and posterior LREs were distinguished by some notable differences in their transcriptional programs. Anterior LREs shared more transcriptional similarities to ileocytes than did posterior LREs. Bile salt transport genes, fatty acid binding protein (*fabp6*), and solute carrier family 10 member 2 (*slc10a2*), were highly expressed in ileocytes, but expression also occurred in the LRE clusters at a gradient from anterior to posterior LREs (*Figure 5C–D*). Further evidence of shared expression patterns between ileocytes and anterior LREs can be seen in their expression of tryptophan metabolic genes. Tryptophan metabolism was a significantly upregulated KEGG pathway in anterior LREs in the GF and CV conditions (*Figure 5—figure supplement 1*). Expression of tryptophan metabolic genes, including kynurenine 3-monoxygenase (*kmo*) and tryptophan 2,3-dioxygenase a (*tdo2a*), occurred in LREs and ileocytes (*Figure 5D–E*). Anterior LREs showed high expression of tryptophan metabolic genes (*tdo2b*, *aldh9a1a.1*, *kmo*, *tdo2a*, *ddc*) in the GF and CV conditions (*Figure 5—figure supplement 1*). Furthermore, anterior and posterior LREs were distinguished by differential expression of several peptidases (*Figure 5—figure supplement 1*).

Several genes were differentially expressed between GF and CV conditions in the anterior and posterior LREs (*Figure 5—figure supplement 1*) (*Supplementary file 5*). The differences between the GF and CV conditions were most apparent in posterior LREs where expression of several peptidase genes was higher in GF than in CV larvae (*Figure 5F*). Carboxypeptidase vitellogenic like (Cpvl), a serine carboxypeptidase, and Cathepsin La (Ctsla), a cysteine-type peptidase, are predicted to localized to lysosomes (*Mahoney et al., 2001*; *Tingaud-Sequeira and Cerdà, 2007*). Peptidase M20 domain containing 1, tandem duplicate 2 (Pm20d1.2) is an amino acid hydrolase (*Long et al., 2016*). These results raise the possibility that higher expression of protein-degradation genes *cpvl*, *ctsla*, and *pm20d1.2* contributes to the faster degradation rates we observed in GF LREs.

In the CV condition, posterior LREs upregulated the expression of several genes involved in the immune response to bacteria (*Figure 5G*). These include genes for several proteins that mediate the immune response to toll-like receptor signaling from bacteria, including *LPS-responsive beige-like anchor protein* (*lrba*), *myeloid differentiation factor 88* (*myd88*), and *hsp90b1* (*Karmarkar and Rock, 2013*; *Gibson et al., 2008*; *Franzenburg et al., 2012*; *van der Sar et al., 2006*; *Graustein et al., 2018*; *Wang et al., 2019*). MyD88 signaling helps to prevent bacterial overgrowth and clears intestinal pathogens (*van der Vaart et al., 2013*; *Karmarkar and Rock, 2013*; *Gibson et al., 2008*; *Franzenburg et al., 2012*; *van der Sar et al., 2006*). MyD88 also regulates expression of *JunD proto-oncogene, AP-1 transcription factor subunit* (*jund*), a LPS-sensor that can induce inflammation in response to the intestinal bacteria, including *Aeromonas* spp. (*Meixner et al., 2004*; *Li et al., 2022*). *EH-domain containing 1b* (*ehd1b*) is upregulated in response to bacterial infection and is involved in

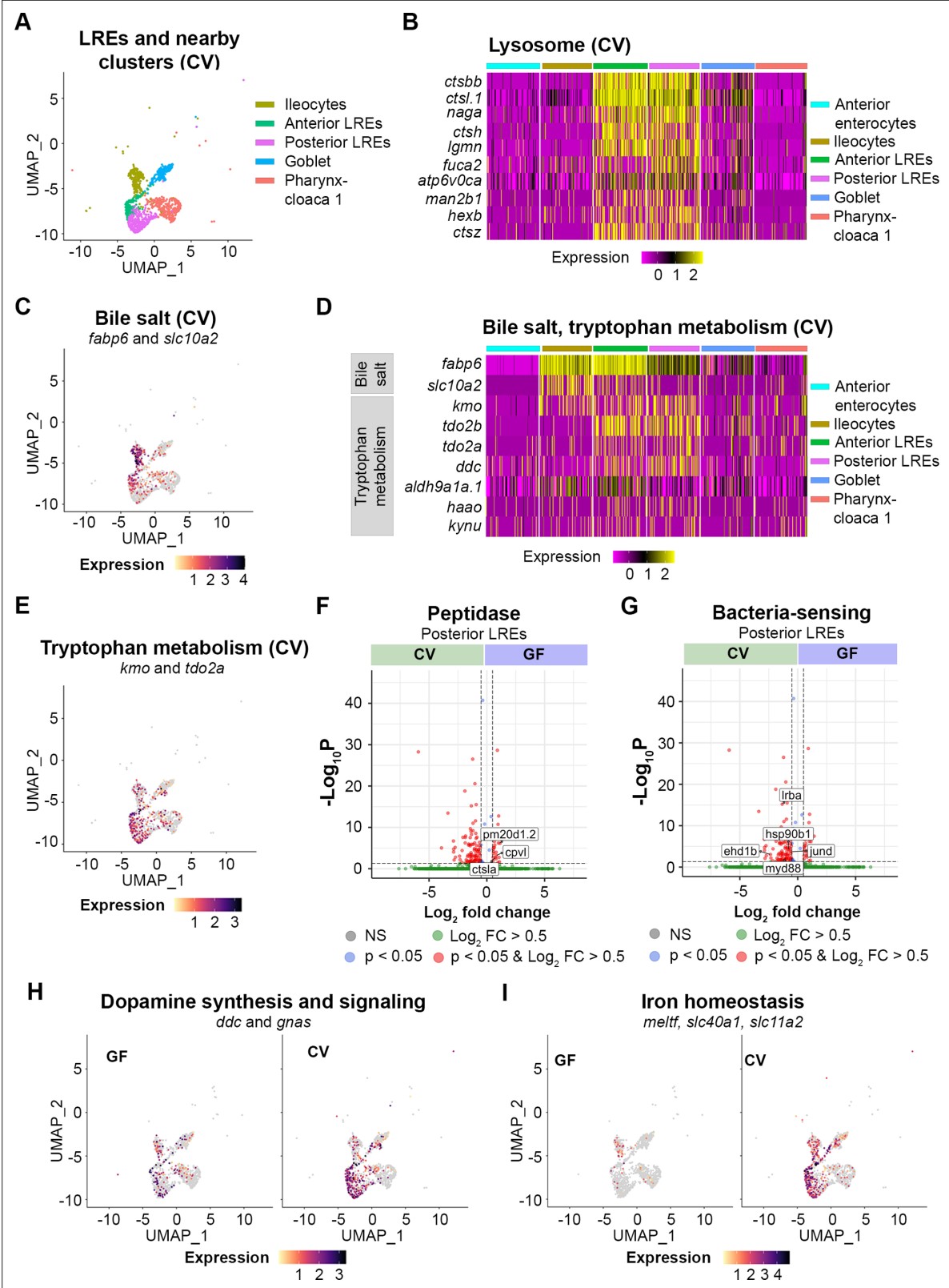

**Figure 5.** Transcriptomic patterns of anterior and posterior lysosome-rich enterocytes (LREs). (**A**) UMAP projection shows anterior and posterior LREs, as well as close cell clusters in the conventional (CV) condition. Cell types are color coded. (**B**) Heatmap illustrates expression of lysosome KEGG pathway genes in LREs and close clusters. The colored bars at the top of the plot indicate the cluster. Heatmap color corresponds to expression intensity. (**C**) UMAP projection displays expression of the bile salt transport genes *fabp6* and *slc10a2* in the LREs, ileocytes, goblet, and pharynx-cloaca 1 cells.

*Figure 5 continued on next page*

*Figure 5 continued*
Cell color indicates cumulative expression intensity for *fabp6* and *slc10a2*. (**D**) Heatmap highlights expression of bile salt transport and tryptophan metabolism genes in the LREs and close clusters. The colored bars at the top indicate the cell cluster. (**E**) UMAP projection displays expression of tryptophan metabolism genes *kmo* and *tdo2a* in the LREs and close clusters. Cell color indicates cumulative expression intensity of *kmo* and *tdo2a*. (**F**) Volcano plot shows differentially expressed genes (DEGs) between germ-free (GF) and CV posterior LREs. Peptidase genes are tagged. (**G**) Volcano plot shows DEGs between GF and CV posterior LREs. Genes involved in microbe sensing and inflammatory response are tagged. (**H**) UMAP projection plots show expression of dopamine synthesis (*ddc*) and signaling (*gnas*) genes in GF (left) and CV (right) cells. (**I**) UMAP projection plots show expression of iron homeostasis genes (*meltf*, *slc40a1*, *slc11a2*) in GF (left) and CV (right) cells.

The online version of this article includes the following figure supplement(s) for figure 5:

**Figure supplement 1.** Lysosome-rich enterocytes (LREs) show regional expression patterns and responses to the gut microbiome.

vesicle-mediated transport (*Dubytska et al., 2022*). These patterns suggest that the microbiome stimulates posterior LREs to upregulate expression of genes involved in directing the immune response to intestinal bacteria.

The microbiome also stimulated expression of dopamine synthesis and signaling genes in posterior LREs (*Figure 5H*). These include *dopamine decarboxylase* (*ddc*), which decarboxylates tryptophan to synthesize dopamine and serotonin (*Koyanagi et al., 2012*). Expression of *GNAS complex locus* (*gnas*), which is involved in the dopamine receptor signaling pathway (*Lu et al., 2006*; *Vortherms et al., 2006*), was also elevated in CV posterior LREs. Interestingly, posterior LREs were the only cell cluster in which *gnas* and *ddc* expression was significantly higher in the CV than GF condition. These results suggest that the gut microbiome increases dopamine synthesis and signaling pathways in posterior LREs.

The microbiome also upregulated several genes related to iron ion transport and homeostasis in LREs (*Figure 5I*). In the CV condition, posterior LREs cells had significantly elevated levels of *solute carrier family 11 member 2* (*slc11a2*), *ferroportin* (*slc40a1*), and *melanotransferrin* (*meltf*) (S5K). Anterior LRE cells also showed increased expression of *slc40a1* and *meltf* (*Figure 5—figure supplement 1*). Slc11a2 helps regulate the influx of iron into the cell, while ferroportin regulates the export of iron out of the cell (*Yilmaz and Li, 2018*; *Bao et al., 2024*). Melanotransferrin has iron-binding properties and is predicted to localize to the plasma membrane, where it transports iron into the cell (*Sekyere et al., 2006*; *Dunn et al., 2007*). Interestingly, melanotransferrin was an important cluster marker in Cloaca 3, a cell cluster that only occurred in the CV condition and is directly adjacent to LREs in the intestine (*Figure 3—figure supplement 1*). These results suggest that upregulation of iron transport genes is a consistent response to microbiome in LREs and other epithelial cells in the distal intestine.

Together, these scRNA-seq data revealed that LREs upregulate genes in response to microbial colonization that are involved in innate immunity, iron homeostasis, and dopamine-synthesis and signaling pathways, and they downregulate several peptidases that may linked to reduced protein degradation upon microbial colonization.

## Individual microbial strains differentially affect LRE activity

We next investigated if specific bacteria alter LRE kinetics and gene expression patterns. To this end, we turned to monoassociation experiments (*Pham et al., 2008*), where larvae are colonized with a single strain of bacteria at 3 dpf rather than the unfractionated microbiome used to colonize CV fish.

We started by testing if commensal bacterial strains isolated from zebrafish gut microbiomes (*Stephens et al., 2016*; *Roeselers et al., 2011*) were sufficient to reduce LRE protein uptake kinetics. To address this question, we measured mCherry uptake in larvae reared either as GF or monoassociated by a single strain, including *Acinetobacter calcoaceticus* ZOR0008, *Aeromonas caviae* ZOR0002, *Vibrio cholerae* ZWU0020, or *Pseudomonas mendocina* ZWU0006 (*Stephens et al., 2016*). At 6 dpf, larvae were gavaged with mCherry (1.25 mg/mL), then mCherry uptake was measured at 1 hr PG (*Figure 6A*). Interestingly, we found that while mCherry uptake was only minimally reduced by *A. calcoaceticus* and moderately reduced by *P. mendocina* and *A. caviae*, colonization with *V. cholerae* reduced mCherry uptake severely (*Figure 6B*).

Next, we investigated the mechanisms by which *V. cholerae* reduced mCherry uptake in LREs. *V. cholerae* monoassociation did not significantly affect fish growth (Two-tailed t-test, p=0.66, n=18–22) (*Figure 6—figure supplement 1*), suggesting that interactions with LREs rather than systemic effects reduced uptake activity. To test if *V. cholerae* exposure induced acute effects on LRE uptake activity,

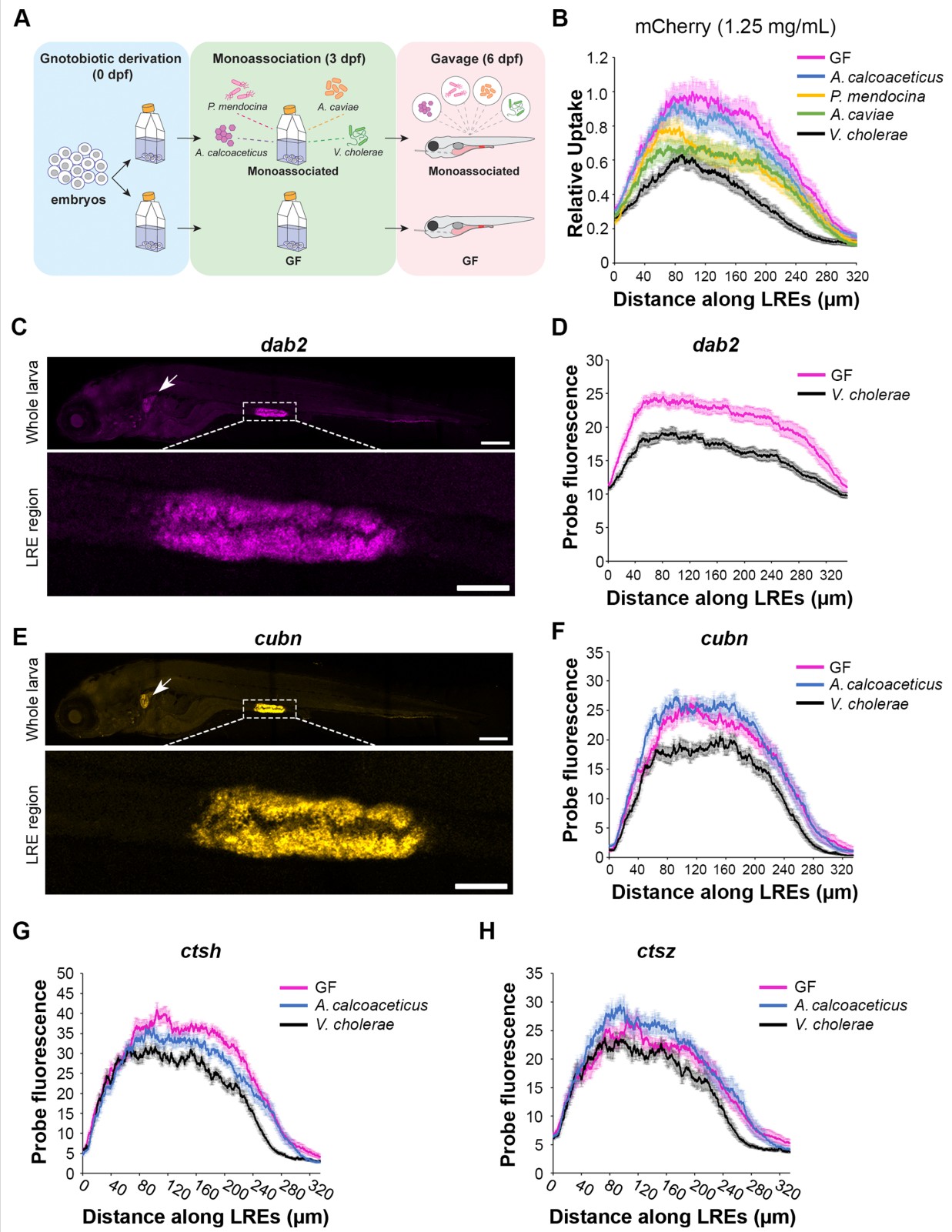

**Figure 6.** Lysosome-rich enterocyte (LRE) activity and expression of endocytic machinery are differentially affected by individual microbial strains. (**A**) Cartoon of monoassociation experimental design. Following gnotobiotic derivation, monoassociated larvae are colonized with a single strain of bacteria at 3 dpf and gavaged with mCherry to measure protein uptake at 6 dpf. (**B**) Plot shows relative uptake of mCherry in larvae that were germ-free (GF) or monoassociated with a single bacterial strain. mCherry uptake was reduced by *A. calcoaceticus* (two-way ANOVA, p=0.0268, n=19–20), *P. mendocina*

*Figure 6 continued on next page*

*Figure 6 continued*

(two-way ANOVA, p=0.0033, n=20–21), *A. caviae* (two-way ANOVA, p<0.0001, n=19–20) and *V. cholerae* (two-way ANOVA, p<0.0001, n=18–20). (**C**) Confocal images show *dab2* hybridization chain reaction (HCR) probe localization in whole zebrafish larva (top) and LRE region (bottom). Arrow points to pronephros. Whole larva scale = 200 µm. LRE region scale = 50 µm. (**D**) Plot shows *dab2* HCR probe fluorescence in the LRE region at 6 dpf. *dab2* expression was significantly greater in GF than *V. cholerae*-colonized larvae (two-way ANOVA, p<0.0001, n=23). (**E**) Confocal images show *cubn* HCR probe localization in whole zebrafish larva (top) and LRE region (bottom). Arrow points to pronephros. Whole larva scale = 200 µm. LRE region scale = 50 µm. (**F**) Plot shows cubn HCR probe fluorescence in the LRE region at 6 dpf. There was greater cubn expression in *A. calcoaceticus*-colonized than GF larvae (two-way ANOVA, p=0.049, n=21–23), but *V. cholerae* significantly reduced cubn expression (two-way ANOVA, p<0.0001, n=16–23). (**G**) Plot of ctsh expression in GF and monoassociated larvae. GF larvae showed greater ctsh expression than A. calcoaceticus (two-way ANOVA, p=0.0183, n=10–12) or *V. cholerae*-colonized larvae (two-way ANOVA, p<0.0001, n=8–12). (**H**) Plot of ctsz expression in GF and monoassociated larvae. A. calcoaceticus and GF larvae showed similar levels of ctsz expression in LREs (two-way ANOVA, p=0.09, n=10–12). *V. cholerae* colonization reduced ctsz expression compared to GF larvae (two-way ANOVA, p=0.0014, n=8–12).

The online version of this article includes the following figure supplement(s) for figure 6:

**Figure supplement 1.** Long-term exposure to *V. cholerae* required to reduce protein uptake activity in lysosome-rich enterocytes (LREs) without affecting larval growth or protein availability.

we gavaged GF larvae with live *V. cholerae* or PBS. Thirty minutes after initial exposure, the larvae were gavaged again with mCherry (1.25 mg per mL). Interestingly, there was no difference in mCherry uptake between larvae that were initially gavaged with live *V. cholerae* or PBS (*Figure 6—figure supplement 1*), indicating that longer-term exposure to *V. cholerae* is required to reduce protein uptake activity in LREs. We next tested if *V. cholerae* monoassociation or CV conditions increased the rate of transcytosis in LREs, which could lead to reduced mCherry signal in LREs. To investigate this possibility, we gavaged GF, CV, and *V. cholerae*-monoassociated larvae with mCherry (1.25 mg/mL) and then measured the mCherry signal in the pronephros at 4 hr PG. There was no difference in pronephros signal between these conditions, indicating that intestinal microbes were not causing LREs to increase mCherry trans-epithelial transport through transcytosis (*Figure 6—figure supplement 1*). Finally, we tested if *V. cholerae* lowered mCherry uptake in LREs by reducing its concentration in the intestinal lumen, perhaps by promoting intestinal motility (*Wiles et al., 2020*; *Logan et al., 2018*). To do so, we analyzed confocal images of GF and *V. cholerae*-colonized larvae taken at 1 hr PG and measured luminal mCherry concentration in the intestinal lumen proximal to LREs. The concentration was the same in both conditions (*Figure 6—figure supplement 1*), showing that *V. cholerae* does not significantly reduce mCherry concentrations in the intestinal lumen within these timeframes.

Our monoassociation experiments suggested that microbes selectively affect LRE activity. To investigate if this effect is mediated by transcriptional regulation of specific protein uptake and degradation machinery genes, we employed quantitative hybridization chain reaction (HCR) RNA-FISH (*Choi et al., 2018*) to compare expression of target genes between GF and monoassociated larvae. We focused on *dab2* and *cubn*, which encode endocytic components critical for the LRE's ability to internalize proteins and soluble cargoes (*Park et al., 2019*).

Similar to what we found previously using conventional in situ hybridization probes (*Park et al., 2019*), the *dab2* HCR probe was highly specific to LREs and pronephros (*Figure 6C*). By comparing the integrated signal obtained from max projection of confocal stacks, we found that GF larvae had significantly higher levels of *dab2* expression than those colonized with *V. cholerae* (*Figure 6D*). Interestingly, the *dab2* expression profiles in GF and *V. cholerae*-colonized larvae showed patterns reminiscent to those of mCherry uptake, with a peak in the anterior LREs that tapers off in the posterior LREs (*Figure 6D*). Since mCherry uptake is dependent on Dab2 (*Park et al., 2019*), this pattern may help explain why mCherry uptake tends to peak in the anterior LREs.

Next, we tested the effects of these strains on *cubn* expression. Similarly to *dab2*, the *cubn* HCR probe highlighted LREs and showed faint fluorescence in pronephros (*Figure 6E*). We found that while *V. cholerae* significantly reduced *cubn* expression, *A. calcoaceticus* did not (*Figure 6F*). These patterns are consistent with the degree to which *V. cholerae* and *A. calcoaceticus* affected mCherry uptake in LREs (*Figure 6B*). Interestingly, the *cubn* expression profile in LREs was different from *dab2*. It lacked an anterior peak, suggesting that *cubn* expression is perhaps more uniform throughout the LREs (*Figure 6F*).

In addition to the endocytic machinery, we also tested the effects of individual microbial strains on the expression of LRE-enriched proteases *ctsh* and *ctsz*. Our scRNA-seq data showed that these

peptidases are highly upregulated in GF LREs. We found that *V. cholerae* reduced expression of *ctsh* and *ctsz* compared to *A. calcoaceticus* and GF larvae, which presented similar levels of *ctsz* expression (*Figure 6G-H*).

Together, our data reveal that microbial colonization reduces both protein uptake and degradation in LREs by modulating the expression of endocytic and lysosomal proteins. Our data also suggest that LREs have functional heterogeneity, which may represent some degree of specialization into two different clusters as suggested by the scRNAseq data.

## LRE activity impacts the gut microbiome in a diet-dependent manner

We next investigated whether LRE activity has reciprocal effects on the gut microbiome. Our previous research demonstrated that *cubn* mutants have significantly reduced survival compared to heterozygotes when they are fed a low protein diet from 6 to 30 dpf (*Park et al., 2019*). We hypothesized that the combination of the *cubn* mutation with a low protein diet fostered a microbial community that further reduced the host's protein uptake capabilities.

To explore this possibility, we tested the combined effects of the *cubn* mutation and custom-formulated, isocaloric high and low protein diets (see Materials and methods) on the larval zebrafish microbiome. We fed high-protein (HP) or low-protein (LP) diets to *cubn* heterozygote and homozygous mutant siblings from 6 to 30 dpf, then performed 16 S rRNA gene sequencing on whole larvae to identify microbial populations (*Figure 7A*). Initial analyses highlighted that zebrafish larvae developed bacterial communities that were distinctly different from the diets and tank water (Bray Curtis distance, p=0.0001) (*Figure 7—figure supplement 1*). Larval microbiomes from different tanks were not significantly different in either HP (Bray Curtis distance, p=0.08) or LP-fed (Bray Curtis distance, p=0.13) conditions (*Figure 7—figure supplement 1*). Diet strongly affected microbiome composition in heterozygotes (Bray Curtis distance, p=0.001), but not in *cubn* homozygous mutants (Bray Curtis distance, p=0.18) (*Figure 7B–C*). Mutants had significantly different microbiomes than heterozygotes when they were fed a LP diet (Bray Curtis distance, p=0.023), but not a HP diet (Bray Curtis distance, p=0.34) (*Figure 7B–C*). In contrast, beta dispersion was not significantly different between HP or LP-fed *cubn* heterozygotes and mutants (Bray Curtis dispersion, p=0.24) (*Figure 7—figure supplement 1*; *Zaneveld et al., 2017*). Some of these effects may reflect differences in taxonomic richness, which was significantly impacted by genotype and diet (1-way ANOVA, p=0.042) (*Figure 7D*). LP-fed *cubn* mutants had significantly lower microbial richness than HP-fed mutants (Two-tailed t-test, p=0.0094, n=6–9). These results suggest that the combination of the LP-diet and *cubn* mutation lowers taxa richness and affects the mutant microbiome.

The broad differences in microbiome diversity led us to investigate class and genus-level effects of the *cubn* mutation and LP diet on the microbiome. The effects of dietary protein on the microbiome were clearly seen at the class level (*Figure 7E*). Gammaproteobacteria were significantly more abundant in HP than LP-fed heterozygotes (DESeq2, padj = 0.02). LP-fed mutants were deficient in Alphaproteobacteria, which were more abundant in LP-fed heterozygotes (DESeq2, padj = 0.021) (*Figure 7E*). Indeed, one important trend was that mutants consistently had fewer differentially abundant taxa than heterozygotes at the class and genus levels (*Figure 7F*). The few taxa that were more abundant in mutants than heterozygotes emerged when they were fed the LP diet (*Figure 7F*).

The impacts of dietary protein and the *cubn* mutation on the microbiome were also apparent at the genus level (*Figure 7F-G*; *Supplementary file 6*). *Pseudomonas* spp. were significantly more abundant in HP than LP-fed heterozygotes (DESeq2, padj = 0.002) (*Figure 7G*). One genus, *Hassalia*, was significantly more abundant in LP-fed mutants than in heterozygotes (DESeq2, padj = 6.04E-21). Independently of diet, *Aeromonas* spp. were more highly abundant in *cubn* mutants than heterozygotes (DESeq2, padj = 0.01) (*Figure 7H*). Interestingly, our monoassociation experiments demonstrated that a member of *Aeromonas* genus, *A. caviae*, can reduce protein uptake activity in LREs (*Figure 6B*).

Together, these results suggest that LRE activity and dietary protein content can interactively affect the larval zebrafish microbiome. The combination of the LP diet and *cubn* mutation cause a less rich microbiome to develop. Furthermore, the *cubn* mutation may lead to the proliferation of certain microbes that can reduce protein uptake activity in LREs.

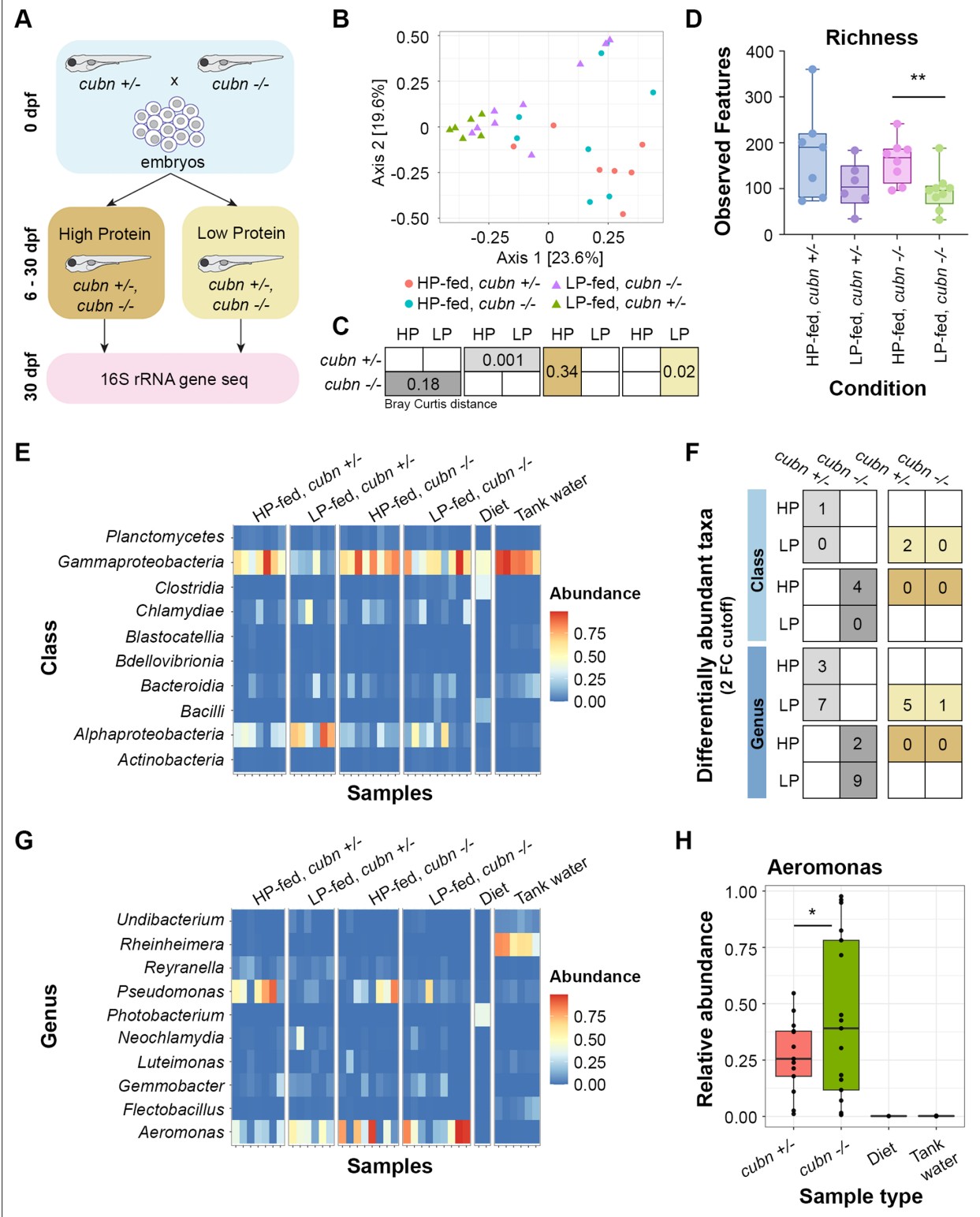

**Figure 7.** Lysosome-rich enterocyte (LRE) activity and dietary protein impact gut microbiome. (**A**) Cartoon of 16 S rRNA gene sequencing experimental design. Homozygous cubn mutant and heterozygote larvae from the same clutch were fed a high-protein (HP) or low-protein (LP) diet from 6 to 30 dpf prior to whole larvae DNA extraction. (**B**) MDS plot of Bray Curtis distance between zebrafish samples. (**C**) Boxes show Bray Curtis distance p-values comparing genotype and diet effects on beta diversity. (**D**) Box plot of observed features between conditions. (**E**) Heat map of classes with the highest relative abundance across all samples. (**F**) Table of differentially abundant taxa counts at the class and genus levels. Boxes show the number of differentially abundant taxa per compared condition. Dietary comparisons are in the left column. Genotype comparisons are in the right column. (**G**)

*Figure 7 continued on next page*

*Figure 7 continued*

Heat map of the genera with the highest relative abundance across all samples. (**H**) Box plot showing relative abundance of Aeromonas spp. across genotypes and controls. The relative abundance of Aeromonas spp. was significantly higher in cubn mutants than heterozygotes (DESeq2, padj = 0.01).

The online version of this article includes the following figure supplement(s) for figure 7:

**Figure supplement 1.** Microbiome assembly shaped by host environment.

## Discussion

The results of our investigations reveal reciprocal interactions between LREs and the gut microbiome that impact host nutrition. We show that the gut microbiome slows the rates of protein uptake and degradation in LREs and reduces expression of lysosomal protease genes. We also found that mono-associated bacteria such as *Aeromonas caviae* and *Vibrio cholerae* reduce protein uptake in LREs and downregulate expression of endocytic and protease genes. Finally, we show that impaired LRE protein uptake activity and dietary protein content impact the microbiome.

Precise delivery of fluorescent cargoes to the gut via gavage provided a quantitative assay that allowed us to uncover that the gut microbiome reduces the rate of cargo uptake and protein degradation in LREs. These effects were caused by microbially-induced metabolic effects in the LREs, not quantity of LREs that developed or the availability of luminal protein. The significance of the microbe-dependent downregulation of protein uptake remains unclear. One possibility is that this effect limits the transepithelial transport of microbial products, which occurs as a byproduct of the endocytic activity of LREs, thus dampening the expression of pro-inflammatory markers. This microbial suppression of the principal physiologic function of LREs (i.e. protein uptake and degradation) is strikingly similar to the microbial suppression of genes involved in lipid absorption in enterocytes within the small intestine of zebrafish (*Camp et al., 2012*; *Willms et al., 2022*) and mice (*Camp et al., 2014*). This suggests that microbial inhibition of absorptive gene expression programs across enterocyte populations may be a broader conserved theme in intestinal physiology.

Recent scRNA-seq studies suggested that the gut microbiome affects the transcriptome of LREs and other ileal cell types in zebrafish by upregulating expression of immune markers (*Massaquoi et al., 2023*; *Jones et al., 2023*; *Willms et al., 2022*). Our dataset also showed that microbes upregulated expression of immune markers in LREs. However, the immune markers that other studies attributed to LREs were not upregulated by LREs in our dataset. For example, we found that several of these immune markers (*saa, prdx1, lect2l*) were actually upregulated in cloaca cells, not LREs (*Willms et al., 2022*). These misalignments could result from previous datasets including relatively few LREs that were over-clustered with cloaca, pronephros, or other ileal cells. Furthermore, these studies did not functionally evaluate intestinal protein absorption, making it difficult to assess the impact of microbes on LRE-dependent processes. Sequencing FACS-sorted IECs minimized contamination from other tissues like the pronephros. Furthermore, we captured a larger number of bona fide LREs from GF and CV larvae that were positively identified in our dataset by their fluorescent label and transcriptional program. Thus, our single-cell dataset provides a high degree of resolution into the effects of the microbiome on LREs and IECs and made it possible to differentiate anterior from posterior LREs.

Labeling intestinal cells by mCherry gavage allowed us to identify all intestinal cells that take up luminal protein, including EECs, goblet, and acinar cells. These cells showed greater expression of LRE markers than mCherry-negative secretory cells. Unlike LREs, they do not appear to play a significant role in protein absorption as gavaging with DQ Red BSA, which labels cells with lysosomal degradation activity, did not label these cells. Remarkably, anterior enterocytes and *best4/otop2* cells were completely mCherry-negative, indicating that they are virtually devoid of endocytic activity.

The proportion of mCherry-positive secretory cells was greater in the CV condition. EECs are known for sensing intestinal nutrients (*Goldspink et al., 2018*), but they also influence the intestinal inflammation response to the gut microbiome (*Worthington et al., 2018*). Intestinal microbes, including *Acinetobacter*, regulate EEC signaling activity in zebrafish larvae (*Ye et al., 2019*). In the mouse intestine, some EECs have a synaptic connection to neurons through neuropods (*Bohórquez et al., 2015*), which could facilitate the uptake of mCherry by neurons that we detected. Neuronal expression of *cldn15la* was detected in our data and other datasets (*Willms et al., 2022*; *Sur et al., 2023*), allowing them to be captured by our cell sorting here. Goblet cells secrete a mucous barrier that protects the intestinal epithelium from microbial infection (*Belkaid and Hand, 2014*). They also sample luminal

antigens and endocytose luminal cargoes as a byproduct of membrane recycling (*Birchenough et al., 2015*; *McDermott and Huffnagle, 2014*). Similarly, acinar cells secrete digestive enzymes and antimicrobial peptides into the intestinal lumen (*Yee et al., 2005*; *Ahuja et al., 2017*). The gut microbiome may stimulate mucin production and the exocytosis of antimicrobial peptides, which would cause these cells to upregulate their compensatory endocytic activity and lead to mCherry uptake. We speculate that the mCherry uptake we observed in these non-LRE cell types could be related to these respective known functions.

The overwhelming majority of LREs were mCherry-positive regardless of microbial colonization. This result aligns with our kinetic assays, which showed that LREs have protein uptake activity in both GF and CV conditions. However, our gavage assays showed that anterior LREs in the GF condition took up protein more rapidly than they did in the CV condition. However, the anterior LREs accumulate more protein than posterior LREs in both the GF and CV conditions. The high uptake activity in this anterior LRE region may be due to greater expression of *dab2*, which our HCR results support. Thus, the anterior LRE region has a greater dynamic range in protein uptake activity, and that may partially explain why the effect of the microbiome was more apparent in this region. In addition, anterior LRE protein uptake activity may have remained more active than posterior LREs in both GF and CV conditions due to reduced expression of innate immune factors compared to posterior LREs. Through high rates of apical endocytosis, LREs could be exposed to large amounts of microbial antigens. Posterior LREs upregulated inflammatory response genes in the CV condition, but anterior LREs did not. Furthermore, anterior LREs showed higher expression of tryptophan metabolism genes than posterior LREs, which may attenuate their inflammatory response to the microbiome. Tryptophan metabolites, including kynurenine derivatives, are AhR receptor ligands that downregulate host immune response to bacterial antigens (*Bessede et al., 2014*). However, tryptophan metabolism may be an important microbiome-tolerance pathway in both LRE regions. The microbiome-induced expression of *haao*, an enzyme that produces kynurenine metabolites (*Xue et al., 2023*), in posterior LREs. These results suggest that tryptophan metabolic activity in LREs could participate in host tolerance of the gut microbiome, and inflammation may affect protein uptake activity in LREs.

LREs may play a role in shaping the gut microbiome community by impacting luminal iron levels. Microbes strongly upregulated expression of iron-ion transport genes in LREs that control the influx and efflux of cellular iron. The microbiome increases intestinal iron absorption (*Mayneris-Perxachs et al., 2022*), but its effect on iron absorption in LREs is unknown. Host iron nutrition and luminal iron levels can affect the gut microbiome community and proliferation of microbial pathogens (*Mayneris-Perxachs et al., 2022*; *Dostal et al., 2012*). Iron absorption and secretion by LREs may affect the gut microbiome community and play a role in host-microbiome homeostasis.

The impact of intestinal microbes on LRE activity may be dependent on their inflammatory properties. Monoassociating larvae with *A. calcoaceticus*, *P. mendocina*, *V. cholerae*, and *A. caviae* produced differential effects on protein uptake activity and expression of endocytic machinery. *A. calcoaceticus* may be classified as an anti-inflammatory bacterial strain because it reduces intestinal neutrophil recruitment (*Rolig et al., 2015*). Colonization with *A. calcoaceticus* only mildly reduced protein uptake activity, and it did not reduce expression of endocytic machinery. In contrast, the pro-inflammatory strains, *V. cholerae* and *A. caviae* (*Rolig et al., 2015*), reduced LRE activity and endocytic expression. *V. cholerae* colonization causes intestinal neutrophil recruitment and stimulates macrophages to express TNF-alpha (*Wiles et al., 2020*; *Ngo et al., 2024*). Importantly, the *V. cholerae* strain we used here does not encode cholera toxin or toxin-coregulated pilus (*Stephens et al., 2016*). This strain has been shown to increase intestinal contractility in zebrafish (Ngo, Amitabh, *Ngo et al., 2024*) and may affect protein absorption under nutrient limiting conditions. However, in our gavage assays the luminal protein concentrations are never limiting and were not affected by *V. cholerae* colonization (Fig. S6D). Future studies could identify the underlying bacterial signals causing LREs to regulate protein uptake activity and host signal transduction mechanisms.

Our investigations into the impact of LRE protein uptake activity on the gut microbiome revealed diet-dependent effects on the microbial community. LP-fed, *cubn* mutants developed a distinct microbiome community with diminished species richness. *Aeromonas* was the only genus that proliferated in *cubn* mutants compared to heterozygotes. While *Aeromonas* is considered a part of the core zebrafish gut microbiome (*Roeselers et al., 2011*), some species are inflammatory (*Rolig et al., 2015*). As discussed, *A. caviae* can reduce LRE protein uptake activity. This result suggests that host

protein deprivation through the impaired LRE activity may lead to proliferation of proinflammatory bacteria like *Aeromonas* that can further reduce protein uptake activity by LREs.

Altogether, our results show reciprocal regulation between LREs and the gut microbiome, along with the impact of the gut microbiome on protein uptake by IECs. These results pave the way to identify microbial signals or antigens that reduce protein uptake activity by LREs. Future studies could characterize the effect of microbes on protein-uptake by secretory cell populations. Further investigations into the tryptophan metabolism, iron ion transport, and expression of bacteria-sensing genes by LREs could reveal mechanisms for host-microbiome homeostasis.

## Methods

### Ethics
Animal experimentation: Zebrafish (*Danio rerio*) were used in accordance with Duke University Institutional Animal Care and Use Committee (IACUC) guidelines and approved under our animal protocol A072-20-03.

### Fish
The Duke University Institutional Animal Care and Use Committee (IACUC) guidelines were followed in the care and use of all fish in this project. We maintained zebrafish (*Danio rerio*) stocks on a recirculating system at 28°C with a controlled, 14-hr light and 10 hr dark cycle (*Westerfield, 2000*). Breeding adult zebrafish were fed a 1:1 ratio of GEMMA Micro (Skretting) and artemia. To breed fish, males and females were placed in mating tanks with dividers overnight, and dividers were removed the following morning. Zebrafish from the Ekkwill (EK) background between 6–30 dpf were used in this study.

### Gnotobiotic zebrafish husbandry
Previously described gnotobiotic husbandry methods were used to raise GF, CV, and monoassociated larvae (*Pham et al., 2008*). Briefly, embryos were treated with antibiotic zebrafish media, followed by iodine and bleach washes to eliminate microbes from their chorions at 0 dpf. Following microbe-removal steps, embryos were housed in sterile cell culture flasks containing autoclaved gnotobiotic zebrafish media (GZM) and incubated at 28 °C. Each flask contained 30 embryos in 30 mL media. At 3 dpf, gnotobiotic larvae were either conventionalized, monoassociated, or remained in the GF condition. In this process, 80% of the media was replaced. The media in GF and monoassociated flasks was replaced with 24 mL autoclaved GZM, while CV flasks received 12 mL autoclaved GZM and 12 mL 5 μm-filtered zebrafish system water. After this step, gnotobiotic zebrafish either (Protocol A) continued to be raised as described (*Pham et al., 2008*), or (Protocol B) they were raised by our modified protocol designed to boost the bacterial load. In Protocol A, each flask was given 100 μL ZM000 daily from 3 to 5 dpf following media changes, and 80% of the media was replaced with autoclaved GZM from 4 to 5 dpf. In Protocol B, flasks are given 150 μL ZM000 from 3 to 5 dpf, and 40% of the media was replaced with autoclaved GZM from 4 to 5 dpf. The scRNA-seq (*Figure 3*, *Figure 4* and *Figure 5*), lucifer yellow gavage (*Figure 1G*), and long-term mCherry gavage (*Figure 1D.E*) larvae were raised according to Protocol A. The larvae used in rapid mCherry-uptake (*Figure 1B,C*), mTurquoise degradation (*Figure 2*), and monoassociation (*Figure 6*) experiments were raised according to Protocol B. At the 6 dpf endpoint, GF flasks were tested for sterility by spot-testing on TSA plates, as well as brain-heart, dextrose and nutrient broth. The bacterial density in CV and monoassociated media was tested by serial dilutions on TSA plates. Zebrafish larvae were monoassociated with *Acinetobacter calcoaceticus* ZOR0008, *Aeromonas caviae* ZOR0002, *Vibrio cholerae* ZWU0020, or *Pseudomonas mendocina* ZWU0006 that were previously isolated from conventionally-reared zebrafish (*Stephens et al., 2016*). Bacteria stocks used in monoassociation were kept in 50% glycerol at –80°C for long-term storage and on tryptic soy agar (TSA) plates at 4 °C for short-term storage. Bacteria were cultured by incubating picked colonies in LB liquid media on a shaker table at 30 °C for 24–72 hr to reach turbidity with an OD600 of 1–3. Cultures were re-suspended in 1 X PBS after spinning at 5000 RPM for 2 min. The bacterial density (CFU/mL) was determined by plating serial dilutions on TSA plates and incubating overnight at 37 °C. Re-suspended bacteria was added to GF flasks to colonize larvae at 3 dpf. For the mCherry gavage experiment (*Figure 6B*), GF larvae were colonized with approximately $6 \times 10^8$ CFU of

*V. cholerae*, *A. caviae*, *P. mendocina*, or *A. calcoaceticus*. The final bacterial density in the media was measured by plating serial dilutions on TSA plates.

## Fluorescent protein purification

Previously described methods were used to prepare mCherry and mTurquoise (*Park et al., 2019*).

## Gavage assays

### Gavage

Larvae were sedated with 0.22 µm-filtered 1X Tricaine (0.2 mg/mL). Sedated larvae were suspended in 3% methyl cellulose and gavaged with 4 nL of fluorescent cargo (*Cocchiaro and Rawls, 2013*). Larvae were placed in a zebrafish incubator to absorb the fluorescent cargo following gavage.

A minimum of twelve larvae were included as biological replicates in each experiment. Individual zebrafish larvae were treated as biological replicates during statistical analysis. Larvae were excluded from analysis if they did not survive the experiment.

### Rapid uptake

Larvae were gavaged with mCherry (1.25 mg/mL), then incubated briefly. After designated time intervals (5–40 min) post gavage (PG), mCherry was cleared from the intestinal lumen by gavaging larvae with 1X PBS. Following clearance, larvae were immediately preserved in 4% PFA in PBS. Samples were stored at 4°C overnight. The following day, larvae were mounted on glass-bottomed dishes and imaged by confocal microscopy. This time course experiment was performed with four groups of larvae in parallel.

Fluorescence profiles of mCherry uptake in LREs were generated by analyzing confocal images in ImageJ (version 1.53t). Images were z-projected as max intensity plots. The LRE region was delineated with a rectangular selection tool to generate plot profiles of mCherry fluorescence along the LRE region. The anterior LRE region was designated as the 50–150 µm region flanking peak fluorescence at 60 min PG. A 100 µm segment length was chosen because it encompasses approximately one-third of the LRE region. In this experiment, the 50–150 µm position was selected because peak fluorescence occurred at approximately 100 µm, so analysis covers the most kinetically active LREs. The posterior LRE region was designated as the 100 µm region immediately distal to the anterior region (150–250 µm).

mCherry fluorescence in anterior versus posterior LREs was calculated from area under the curve (AUC) measurements (*Figure 1—figure supplement 1*). The average AUC measure at each time point was calculated. These values were normalized by dividing them by the peak AUC value at 60 min post gavage in the GF condition. The regional differences in mCherry fluorescence between anterior and posterior LREs from 5–60 min post gavage was calculated with a two-way ANOVA with Bonferroni's multiple comparisons.

The rate of mCherry accumulation in anterior LREs (*Figure 1—figure supplement 1*) was calculated next. The average and standard error in mCherry fluorescence at each time point was calculated for GF and CV conditions. These were normalized when they were divided by the maximum average value, which occurred in the GF condition at 60 min PG. A simple linear regression was used to calculate the difference in the mCherry-accumulation slopes between GF and CV conditions.

### Long-term uptake of mCherry (1–5 hr PG)

Following mCherry gavage (1.25 mg/mL), larvae were incubated, then preserved in 4% PFA at designated time intervals (1–5 hr PG). Samples were stored at 4°C overnight, then imaged by confocal microscopy. Confocal images were analyzed with ImageJ. The same protocol was used to process images. To compare mCherry and Lucifer yellow uptake in the whole LRE region, the average fluorescence and standard error were calculated for each 30 µm LRE segment in GF and CV conditions (*Figure 1D–F*). The difference in fluorescence profiles between GF and CV conditions at each time point was then calculated with a two-way ANOVA. To do so, the average fluorescence in each segment (30 µm) of the LRE region (300 µm) was compared between the GF and CV conditions.

To compare the rate of mCherry uptake in anterior LREs, the average fluorescence and standard error were calculated for the 0–100 µm segment in GF and CV conditions. A simple linear regression calculated the difference in mCherry-uptake rate between conditions, along with the difference in

slope from zero. A 100 µm segment length was chosen because it encompasses approximately one-third of the LRE region. In this experiment, the 0–100 µm position was selected because peak fluorescence occurred at approximately 50 µm, so analysis covered the most kinetically active LREs. This experiment was replicated at least three times (data not shown) with one representative experiment depicted in *Figure 1*.

## Lucifer yellow uptake

Larvae were gavaged with 4 nL of Lucifer yellow (1.25 mg/mL), then incubated. Immediately before imaging, luminal Lucifer yellow was cleared with a 1X PBS gavage. Larvae were live imaged to avoid signal quenching by PFA. The same protocol was used to process these images as mCherry-gavaged larvae. To compare Lucifer yellow uptake in the whole LRE region, the average fluorescence and standard error were calculated for each 50 µm LRE segment in GF and CV conditions. The difference in fluorescence between GF and CV conditions at each time point was then calculated with a two-way ANOVA. The same statistical methods were used to calculate the two-way ANOVA as the long-term mCherry uptake experiments. This experiment was replicated three times (data not shown) with one representative experiment depicted in *Figure 1*.

## mTurquoise degradation

Larvae were gavaged with mTurquoise (25 mg/mL) and incubated for 1 hr. Luminal mTurquoise was cleared with a 1X PBS gavage. Sedated larvae were mounted on glass-bottom plates and live imaged by confocal microscopy from 20–60 min post clearance.

mTurquoise fluorescence was quantified in the whole LRE region using ImageJ as described above. Live imaging allowed us to track mTurquoise degradation over time in individual fish. The maximum mTurquoise fluorescence value for each individual fish was calculated at the first time point (20 min). All mTurquoise fluorescence measures for each respective fish were then divided by the fish's maximum value, effectively converting mTurquoise fluorescence measures to a 0–100% scale across all time points. After that, we calculated average mTurquoise fluorescence in individual fish at each time point and normalized to the fish's average value at 20 min PG. Next, we generated a degradation curve by finding the mean and standard error across fish from GF or CV conditions. These values were linearized when we transformed the x-axis by dividing 1 by the time point (i.e. the x-axis value for the 20-min time point would be 1/0.333 or 3). Finally, we calculated the difference in degradation rate between GF and CV larvae with a simple linear regression. This experiment depicted in *Figure 2* was performed one time with live larval samples, and it was replicated with fixed larvae (data not shown).

## Microbial degradation of mCherry

We tested if the larval zebrafish microbiome can degrade mCherry. CV larvae (n = 27) were anesthetized with 0.22 µm-filtered Tricaine, then homogenized in GZM with a Tissue-Tearor (BioSpec Products, inc, model # 985370). The homogenate was spun at 5000 RPM for 2 min and re-suspended in 1X PBS. mCherry was added to the zebrafish microbiome mixture and the 1X PBS control (25 µg/mL). The microbiome and control were added to a 96-well plate. The mCherry fluorescence was measured in each well periodically over 2 hr. Average mCherry fluorescence (AU) over time was compared between treatments with a simple linear regression. This experiment was performed with 9-10 biological replicates for GF and CV conditions.

## Trans-epithelial transport

GF and CV larvae were anesthetized with 1X Tricaine at 6 dpf. Then, anesthetized larvae were gavaged with mCherry protein (1.25 mg/mL). Larvae were placed in a 28°C incubator to absorb the mCherry for 4 hr. At that point, larvae were fixed in 4% PFA and stored overnight at 4°C. Fixed samples were washed in 1X PBS three times before being mounted in 0.9% low-melt agarose in egg water. Imaging was done with a 25X objective and resonant scanner, and images were taken as z-stacks. Statistics were done by one-way ANOVA with Tukey's multiple comparisons test. This experiment was replicated in GF and CV larvae two times with one representative experiment shown (*Figure 6—figure supplement 1*). Results were replicated when comparing trans-epithelial transport between GF, CV, and *V. cholerae*-colonized larvae (data not shown).

## LRE segmentation

GF and CV larvae were anesthetized with 1X Tricaine at 6 dpf. Then, DQ-red BSA (50 µg/mL) was gavaged into larvae until the whole gut was filled with the gavage mixture (4 nL). After 0.5 hr of uptake, the larvae were flushed with a 1X PBS gavage until no visible red color was observed in the gut. Larvae were stored in a 28°C incubator for 3 hr after DQ-red BSA gavage. During this incubation period, DQ-red BSA fluorescence activated when the dye was catabolized by lysosomal proteases, thus marking the lysosomal vacuole of each LRE. Larvae were then anesthetized with 1X Tricaine, mounted in 1.3% low-melt agarose in egg water, and live imaged by confocal microscopy. Imaging was done on a Leica SP8 microscope using a 25X objective (Leica) and resonant scanner.

Images were imported into Ilastik version 1.3.3 and segmented by the Pixel Classification program. The program is trained to identify DQ-red BSA-filled LRE vacuoles by machine learning where 10–15 LRE vacuoles are manually marked per image. This process is carried out for 5–10 images before the program is allowed to train itself based on these manual feed-ins. Any errors made by the program are then manually corrected and used to improve the segmentation process. After the segmentation program is trained to have at least an estimated 85% accuracy, the results are exported as probability maps in.h5 format, which are then fed into Ilastik's Object Classification program. The following parameters are used for the classification process: Method: Hysteresis; Smooth: σ=1.0 for x-,y- and z-axis; Threshold: core = 0.60, final=0.60; Don't merge objects. After checking the accuracy of object classification, the results are then exported as.cvs files from which the numbers of objects (in this case LRE vacuoles) are extracted. Statistics were done by 1-way ANOVA with Tukey's multiple comparisons test. Quantification was done with 6–8 biological replicates from the GF and CV conditions in this experiment.

## Fluorescence activated cell sorting for single-cell RNA-sequencing

*TgBAC(cldn15la-GFP)*[pd1034] larvae (*Alvers et al., 2014*) were raised in GF and CV conditions to 6 dpf as described above. These transgenic larvae express GFP in IECs. GF flasks were screened for sterility as described above. Bacterial density in CV flasks ranged from approximately $5 \times 10^4$ – $5 \times 10^9$ CFU/mL. In total, 276 CV and 277 GF, GFP-positive larvae were gavaged with mCherry (1.25 mg/mL) and incubated for 4 hr to mark LREs with mCherry in addition to GFP. Then, larvae were dissociated, and cell suspensions were sorted by FACS as described (*Park et al., 2019*). The cell death marker, 7AAD (5 µg/mL), was added to cell suspension prior to FACS to remove dead cells. Cells were sorted on a MoFlo Astrios EQ cell sorter (Beckman Coulter) by the Flow Cytometry Shared Resource Center (Duke University). In total, 61,000 double-positive and 156,000 GFP-positive cells were collected from GF larval samples. From CV larval samples, 51,000 double-positive and 140,000 GFP-positive cells were collected. Cells were collected in RLT Plus Buffer (Qiagen RNeasy Plus Micro Kit: QIAGEN Cat No. 74034) media.

To test mCherry-uptake activity in this cohort, GF and CV larvae were gavaged with mCherry (1.25 mg/mL) at the same time as larvae destined for scRNA-seq. These larvae were incubated for 5 hr, then preserved in 4% PFA. They were confocal imaged the following day as described above.

## Single-cell RNA-sequencing library preparation and sequencing

Immediately after cells were sorted, we began library preparation with the Chromium Next GEM Single Cell 3' GEM, Library & Gel Bead Kit v3.1 following the kit protocols. GEMS were stored at –20°C for 3 d prior to post-GEM-RT cleanup and cDNA amplification following the kit protocol. Quality control was done by measuring cDNA concentration and size distribution with the ScreenTape assay using the Agilent TapeStation. Libraries were submitted to the Duke Center for Genomic and Computational Biology for sequencing. They were sequenced on the Illumina NovaSeq 6000 platform with 100 bp paired-end reads. Sequencing results are deposited at NCBI Sequence Read Archive (BioProject Accession #: PRJNA1192682).

## Single-cell RNA-sequencing analysis

Cell Ranger v3.0 (10X Genomics) was used to demultiplex sequencing files and align reads to the zebrafish reference genome, *Danio rerio* GRCz11. Analysis was performed in RStudio (version 2023.12.1) using the Seurat package (version 5.0.1) (*Hao et al., 2024*). Reads were filtered by the following criteria: UMIs/cell >500, genes/cell >250, Log10GenesperUMI >0.8, and mitoRatio <0.5.

The SCTransform method was used for normalization. Cells were clustered with a resolution of 0.5. Cluster markers were identified using Seurat's 'FindMarkers' function. Cluster identity was determined by cross-referencing marker genes to other scRNA-seq and RNA-sequencing datasets (*Park et al., 2019*; *Wen et al., 2021*; *Willms et al., 2022*). After the LRE cluster was identified, it was re-clustered with a resolution of 0.5 to form two sub-clusters. The GF and CV objects were merged to form one Seurat object.

## In situ hybridization chain reaction (HCR)

HCR probes, hairpins, amplification, wash, and hybridization buffers were purchased from Molecular Instruments (*Choi et al., 2018*). Our methods were adapted from a previously published procedure for performing HCR on zebrafish embryos (*Munjal et al., 2021*). At 6dpf, larvae were anesthetized and fixed in 4% PFA, then incubated on a shaker table for 2 hr. Larvae were washed with 1X PBS twice, followed by two cold acetone washes. Larvae were incubated in cold acetone at –20°C for 8 min, followed by three 1X PBS washes. For the detection stage, larvae were first incubated in probe hybridization buffer at 37°C for 30 min on a shaker. Larvae were incubated in probe solution (4 nM) on a shaker at 37 °C for 24 – 48 hr. Excess probes were removed by washing larvae four times with preheated, 37°C probe wash buffer, incubating samples on a shaker table at room temperature for 15 min each time. This step was followed by two, 5–10 min SSCT washes on a shaker at room temperature. For the amplification stage, larvae were incubated in room temperature amplification buffer for 30 min. Hairpins (30 pmol) were prepared by heating at 95°C for 90 s, then snap-cooled in the dark at room temperature 30 min. Hairpins were added to amplification buffer. Larvae were incubated in the hairpin solution in the dark at room temperature overnight. Excess hairpins were removed with five SSCT washes on a shaker table at room temperature. The duration of the first two washes were 5 min, followed by two 30-min washes and one 1-min wash. Samples were protected from light during the washes. Larvae were imaged by confocal microscopy using a Leica SP8 microscope equipped with 10X and 25 X objectives. The difference in HCR probe fluorescence profiles between GF and mono-associated larvae was calculated with a two-way ANOVA. To do so, the average probe fluorescence in each segment (30 μm) of the LRE region (300 μm) was compared between the GF and monoassociated conditions. HCR probe sequences are listed in *Supplementary file 7*. HCR probes were tested and imaged in at least three replicate experiments (data not shown). HCR probe fluorescence was measured in GF and monoassociated larvae in two replicate experiments and multiple larvae as indicated in the figure legend.

## High and low-protein diet feeding

Sibling larvae from a *cubn* heterozygous-mutant cross were conventionally raised from 0–5 dpf. At 6 dpf larvae were housed in 3 L tanks at a density of 10 larvae per tank and raised on our standard circulating aquarium. From 6 – 30 dpf, larvae were fed 10 mg/d of a custom-formulated high (HP) or low-protein (LP) diet daily between 11 am–12 pm. There were three tanks per diet. Diet formulations are described in detail below.

## 16S rRNA gene sequencing

At 30 dpf, samples were collected for 16S rRNA gene sequencing. Larvae were anesthetized with 0.2 μm-filtered tricaine in autoclaved egg water. After the water was removed, larvae were flash frozen in liquid nitrogen. Tank water samples were collected by passing 50 mL through a 0.2 μm filter (Pall Corporation MicroFunnel Filter Funnels #4803), then flash freezing the filter paper in liquid nitrogen.

Samples were put on dry ice during transport to the Duke Microbiome Shared Resource for 16S rRNA gene sequencing. Larval DNA was extracted with the MagAttract PowerSoil DNA EP Kit (Qiagen, 27100-4-EP). Following the Earth Microbiome Project protocol (http://www.earthmicrobiome.org/), the V4 region was amplified by polymerase chain reaction with the 515F and 806R primers, which are barcoded for multiplexed sequencing. PCR product concentration was measured with the Qubit dsDNA HS assay kit (ThermoFisher, Q32854) on a Promega GloMax plate reader. Equimolar PCR products from all samples were pooled and sequenced with 250 bp PE reads on the MiSeq Illumina platform. Sequencing results are deposited at NCBI Sequence Read Archive (BioProject Accession #: PRJNA1188138).

Paired-end fastq files were demultiplexed in R (version 4.1.1). The *dada2* package was used to denoise the sequences, filter, and trim reads, dereplicate reads, merge paired end reads, generate an amplicon sequence variant table, remove chimeras, and generate a Phyloseq object (*Callahan et al., 2016a*; *Callahan et al., 2016b*). The Silva database (version 138.1) was used to assign taxonomy. Mitochondria and chloroplasts were filtered out of the dataset using the Phyloseq package (version 1.46.0) (*McMurdie and Holmes, 2013*). Phyloseq was used for downstream analysis of relative abundance and taxa richness. The DESeq2 package (version 1.42.0) was used to measure differential abundance (*Love et al., 2014*). The ggplot2 package (version 3.5.0) was used to generate heatmap and relative abundance plots. The vegan package (version 2.6-4) was used to measure Bray Curtis distance.

The following primers were used to genotype the larvae: cubn F: 5'-ACTCTGTTCACCTGCAGTGC-3', cubn R: 5'-TGACATCCGAGTGGAGTTCCTGCCAAGAC-3'.

## Diet formulations

These diets were custom-formulated at University of Alabama at Birmingham.

| | Z17-D01 | Z17-D02 |
|---|---|---|
| | High protein | Low protein |
| Ingredient | % | % |
| Casein - low trace metals | 30.00 | 15.00 |
| Fish protein hydrolysate | 27.00 | 13.50 |
| Alpha cellulose | 8.00 | 8.00 |
| Wheat starch | 5.09 | 23.00 |
| Dextrin | 4.00 | 4.00 |
| Safflower oil | 4.00 | 5.00 |
| Soy lecithin (refined) | 4.00 | 4.00 |
| Vitamin mix (MP-VDFM) | 4.00 | 4.00 |
| Diatomaceous earth | 3.20 | 12.29 |
| Aalginate (TIC algin 400) | 3.00 | 3.00 |
| Mineral mix (BTm) | 3.00 | 3.00 |
| Menhaden fish oil (ARBP) | 2.00 | 2.50 |
| Potassium phosphate monobasic | 1.15 | 1.15 |
| Canthaxanthin (10%) | 1.00 | 1.00 |
| Glucosamine | 0.25 | 0.25 |
| Betaine | 0.15 | 0.15 |
| Cholesterol | 0.12 | 0.12 |
| Ascorbylpalmitate | 0.04 | 0.04 |
| Total | 100.00 | 100.00 |
| Calculated Protein (%) | 50.10 | 25.05 |
| Calculated Fat (%) | 13.12 | 13.12 |
| Calculated Carbohydrate (%) | 12.41 | 30.32 |
| Calculated Energy (cal/g) | 4567 | 3868 |

## Acknowledgements

We thank Dan Levic and Carina Block for critical reading of our manuscript, Colin Lickwar for helpful advice on bioinformatic analysis, Jia Wen for advice on monoassociation, Akankshi Munjal for advice

on HCR, the Duke ZeCore for fish care, Duke Microbiome Core for assistance with 16 S rRNA gene sequencing, and the Duke Center for Genomic and Computational Biology for help with scRNA sequencing. This work was funded by NIH grants DK132120 to MB and DK121007 to MB and JFR.

## Additional information

### Competing interests
Michel Bagnat: Reviewing editor, *eLife*. The other authors declare that no competing interests exist.

### Funding

| Funder | Grant reference number | Author |
|---|---|---|
| National Institutes of Health | DK132120 | Michel Bagnat |
| National Institutes of Health | DK121007 | John F Rawls<br>Michel Bagnat |

The funders had no role in study design, data collection and interpretation, or the decision to submit the work for publication.

### Author contributions
Laura Childers, Formal analysis, Investigation, Visualization, Methodology, Writing - original draft, Writing – review and editing; Jieun Park, Conceptualization, Formal analysis, Investigation, Methodology, Writing – review and editing; Siyao Wang, Investigation, Methodology; Richard Liu, Investigation; Robert Barry, Resources; Stephen A Watts, Resources, Writing – review and editing; John F Rawls, Conceptualization, Resources, Formal analysis, Supervision, Funding acquisition, Investigation, Methodology, Writing – review and editing; Michel Bagnat, Conceptualization, Resources, Data curation, Formal analysis, Supervision, Funding acquisition, Investigation, Methodology, Writing - original draft, Project administration, Writing – review and editing

### Author ORCIDs
Laura Childers ⓘ https://orcid.org/0000-0003-0761-2256
John F Rawls ⓘ https://orcid.org/0000-0002-5976-5206
Michel Bagnat ⓘ https://orcid.org/0000-0002-3829-0168

### Ethics
The Duke University Institutional Animal Care and Use Committee (IACUC) guidelines were followed in the care and use of all fish in this project. Protocol A072-20-03.

Reviewer #1 (Public review): https://doi.org/10.7554/eLife.100611.3.sa1
Reviewer #2 (Public review): https://doi.org/10.7554/eLife.100611.3.sa2
Reviewer #3 (Public review): https://doi.org/10.7554/eLife.100611.3.sa3
Author response https://doi.org/10.7554/eLife.100611.3.sa4

## Additional files

### Supplementary files
Supplementary file 1. Cluster markers in the GF dataset.

Supplementary file 2. Cluster markers in the CV dataset.

Supplementary file 3. Gene expression levels across clusters.

Supplementary file 4. Differentially expressed genes between mCherry-positive and mCherry-negative cells.

Supplementary file 5. Differentially expressed genes between germ-free (GF) and conventional (CV) conditions.

Supplementary file 6. Differentially abundant bacteria.

Supplementary file 7. Hybridization chain reaction (HCR) probe sequences.

MDAR checklist

## Data availability

Single cell RNA-sequencing data has been deposited at the NCBI Sequence Read Archive (BioProject Accession #: PRJNA1192682). The 16S rRNA gene sequencing data has been deposited at the NCBI Sequence Read Archive (BioProject Accession #: PRJNA1188138). Code used in 16S rRNA gene sequencing and single-cell RNA sequencing analysis is available on GitHub (copy archived at *Childers, 2025*).

The following datasets were generated:

| Author(s) | Year | Dataset title | Dataset URL | Database and Identifier |
|---|---|---|---|---|
| Childers L, Park J, Wang S, Liu R, Barry R, Watts S, Rawls JF, Bagnat M | 2024 | Microbial regulation of gene expression patterns in lysosome rich enterocyte (LRE) and intestinal epithelial cells | https://www.ncbi.nlm.nih.gov/bioproject/PRJNA1192682 | NCBI Bioproject, PRJNA1192682 |
| Childers L, Park J, Wang S, Liu R, Barry R, Watts S, Rawls JF, Bagnat M | 2024 | Dietary protein and lysosome rich enterocyte (LRE) activity impact zebrafish microbiome | https://www.ncbi.nlm.nih.gov/bioproject/PRJNA1188138 | NCBI Bioproject, PRJNA1188138 |

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
