## [Editor Report · eLife Assessment]

In this study, the authors use the zebrafish to investigate how the microbiome affects a specialized gut cell called the lysosome rich enterocyte. They use a combination of functional assays for protein absorption, gnotobiotic manipulations and single-cell RNA-seq. The findings in the paper are considered **important** and the results are **convincing**.

---

## [Referee Report · Reviewer #1 (Public review)]

The Bagnat and Rawls groups' previous published work (Park et al., 2019) described the kinetics and genetic basis of protein absorption in a specialized cell population of young vertebrates termed lysosome rich enterocytes (LREs). In this study they seek to understand how the presence and composition of the microbiota impacts the protein absorption function of these cells and reciprocally, how diet and intestinal protein absorption function impact the microbiome.

Strengths of the study include the functional assays for protein absorption performed in live larval zebrafish, which provides detailed kinetics on protein uptake and degradation with anatomic precision, and the gnotobiotic manipulations. The authors clearly show that the presence of the microbiota or of certain individual bacterial members slows the uptake and degradation of multiple different tester fluorescent proteins.

To understand the mechanistic basis for these differences, the authors also provide detailed single-cell transcriptomic analyses of cells isolated based on both an intestinal epithelial cell identity (based on a transgenic marker) and their protein uptake activity. The data generated from these analyses, presented in Figures 3-5, are valuable for expanding knowledge about zebrafish intestinal epithelial cell identities, but of more limited interest to a broader readership. Some of the descriptive analysis in this section is circular because the authors define subsets of LREs (termed anterior and posterior) based on their fabp6 expression levels, but then go on to note transcriptional differences between these cells (for example in fabp6) that are a consequence of this initial subsetting.

Inspired by their single-cell profiling and by previous characterization of the genes required for protein uptake and degradation in the LREs, the authors use quantitative hybridization chain reaction RNA-fluorescent in situ hybridization to examine transcript levels of several of these genes along the length of the LRE intestinal region of germ-free versus mono-associated larvae. They provide good evidence for reduced transcript levels of these genes that correlate with the reduced protein uptake in the mono-associated larval groups.

The final part of the study (shown in Figure 7) characterized the microbiomes of 30-day-old zebrafish reared from 6-30 days on defined diets of low and high protein and with or without homozygous loss of the cubn gene required for protein uptake. The analysis of these microbiomes notes some significant differences between fish genotypes by diet treatments, but the discussion of these data does not provide strong support for the hypothesis that "LRE activity has reciprocal effects on the gut microbiome". The most striking feature of the MDS plot of Bray Curtis distance between zebrafish samples shown in Figure 7B is the separation by diet independent of host genotype, which is not discussed in the associated text. Additionally, the high protein diet microbiomes have a greater spread than those of the low protein treatment groups, with the high protein diet cubn mutant samples being the most dispersed. This pattern is consistent with the intestinal microbiota under a high protein diet regimen and in the absence of protein absorption machinery being most perturbed in stochastic ways than in hosts competent for protein uptake, consistent with greater beta dispersal associated with more dysbiotic microbiomes (described as the Anna Karenina principle here: https://pubmed.ncbi.nlm.nih.gov/28836573/). It would be useful for the authors to provide statistics on the beta dispersal of each treatment group.

Overall, this study provides strong evidence that specific members of the microbiota differentially impact gene expression and cellular activities of enterocyte protein uptake and degradation, findings that have a significant impact on the field of gastrointestinal physiology. The work refines our understanding of intestinal cell types that contribute to protein uptake and their respective transcriptomes. The work also provides some evidence that microbiomes are modulated by enterocyte protein uptake capacity in a diet-dependent manner. These latter findings provide valuable datasets for future related studies.

Comments on revisions:

I suggest that the authors clarify the level of protein in the standard fish food and how this relates to the protein levels in the high protein and low protein diets used in their microbiome study.

---

## [Referee Report · Reviewer #2 (Public review)]

Summary:

The authors set out to determine how the microbiome and host genotype impact host protein-based nutrition.

Strengths:

The quantification of protein uptake dynamics is a major strength of this work and the sensitivity of this assay shows that the microbiome and even mono-associated bacterial strains dampen protein uptake in the host by causing down-regulation of genes involved in this process rather than a change in cell type.

The use of fluorescent proteins in combination with transcript clustering in the single cell seq analysis deepens our understanding of the cells that participate in protein uptake along the intestine. In addition to the lysozome-rich enterocytes (LRE), subsets of enteroendocrine cells, acinar, and goblet cells also take up protein. Intriguingly, these non-LRE cells did not show lysosomal-based protein degradation; but importantly analysis of the transcripts upregulated in these cells include dab2 and cubn, genes shown previously as being essential to protein uptake.

The derivation of zebrafish mono-associated with single strains of microbes paired with HCR to localize and quantify the expression of host protein absorption genes shows that different bacterial strains suppress these genes to variable extents.

The analysis of microbiome composition, when host protein absorption is compromised in cubn-/- larvae or by reducing protein in the food, demonstrates that changes to host uptake can alter the abundance of specific microbial taxa like Aeramonas.

---

## [Referee Report · Reviewer #3 (Public review)]

Childers et al. address a fundamental question about the complex relationship within the gut: the link between nutrient absorption, microbial presence, and intestinal physiology. They focus on the role of lysosome-rich enterocytes (LREs) and the microbiota in protein absorption within the intestinal epithelium. By using germ-free and conventional zebrafishes, they demonstrate that microbial association leads to a reduction in protein uptake by LREs. Through impressive in vivo imaging of gavaged fluorescent proteins, they detail the degradation rate within the LRE region, positioning these cells as key players in the process. Additionally, the authors map protein absorption in the gut using single-cell sequencing analysis, extensively describing LRE subpopulations in terms of clustering and transcriptomic patterns. They further explore the monoassociation of ex-germ-free animals with specific bacterial strains, revealing that the reduction in protein absorption in the LRE region is strain-specific.

Strengths:

- The authors employ state-of-the-art imaging to provide clear evidence of the protein absorption rate phenotype, focusing on a specific intestinal region. This innovative method of fluorescent protein tracing expands the field of in vivo gut physiology.

- Using both conventional and germ-free animals for single-cell sequencing analysis, they offer valuable epithelial datasets for researchers studying host-microbe interactions. By capitalizing on fluorescently labelled proteins in vivo, they create a new and specific atlas of cells involved in protein absorption, along with a detailed LRE single-cell transcriptomic dataset.

- Their robust and convincing microbiota analysis puts forward a diet-dependent mechanism of community change upon low-protein diet, intricately linked with the host.

Comments on revisions:

The authors have improved the manuscript following the revision work. No further recommendations.

---

## [Author Response]

The following is the authors’ response to the original reviews.

**Public Reviews:**

**Reviewer #1 (Public review):**
The Bagnat and Rawls groups' previous published work (Park et al., 2019) described the kinetics and genetic basis of protein absorption in a specialized cell population of young vertebrates termed lysosome-rich enterocytes (LREs). In this study they seek to understand how the presence and composition of the microbiota impacts the protein absorption function of these cells and reciprocally, how diet and intestinal protein absorption function impact the microbiome.Strengths of the study include the functional assays for protein absorption performed in live larval zebrafish, which provides detailed kinetics on protein uptake and degradation with anatomic precision, and the gnotobiotic manipulations. The authors clearly show that the presence of the microbiota or of certain individual bacterial members slows the uptake and degradation of multiple different tester fluorescent proteins.To understand the mechanistic basis for these differences, the authors also provide detailed single-cell transcriptomic analyses of cells isolated based on both an intestinal epithelial cell identity (based on a transgenic marker) and their protein uptake activity. The data generated from these analyses, presented in Figures 3-5, are valuable for expanding knowledge about zebrafish intestinal epithelial cell identities, but of more limited interest to a broader readership. Some of the descriptive analysis in this section is circular because the authors define subsets of LREs (termed anterior and posterior) based on their fabp2 expression levels, but then go on to note transcriptional differences between these cells (for example in fabp2) that are a consequence of this initial subsetting.Inspired by their single-cell profiling and by previous characterization of the genes required for protein uptake and degradation in the LREs, the authors use quantitative hybridization chain reaction RNA-fluorescent in situ hybridization to examine transcript levels of several of these genes along the length of the LRE intestinal region of germ-free versus mono-associated larvae. They provide good evidence for reduced transcript levels of these genes that correlate with the reduced protein uptake in the mono-associated larval groups.The final part of the study (shown in Figure 7) characterized the microbiomes of 30-day-old zebrafish reared from 6-30 days on defined diets of low and high protein and with or without homozygous loss of the cubn gene required for protein uptake. The analysis of these microbiomes notes some significant differences between fish genotypes by diet treatments, but the discussion of these data does not provide strong support for the hypothesis that "LRE activity has reciprocal effects on the gut microbiome". The most striking feature of the MDS plot of Bray Curtis distance between zebrafish samples shown in Figure 7B is the separation by diet independent of host genotype, which is not discussed in the associated text. Additionally, the high protein diet microbiomes have a greater spread than those of the low protein treatment groups, with the high protein diet cubn mutant samples being the most dispersed. This pattern is consistent with the intestinal microbiota under a high protein diet regimen and in the absence of protein absorption machinery being most perturbed in stochastic ways than in hosts competent for protein uptake, consistent with greater beta dispersal associated with more dysbiotic microbiomes (described as the Anna Karenina principle here: https://pubmed.ncbi.nlm.nih.gov/28836573/). It would be useful for the authors to provide statistics on the beta dispersal of each treatment group.Overall, this study provides strong evidence that specific members of the microbiota differentially impact gene expression and cellular activities of enterocyte protein uptake and degradation, findings that have a significant impact on the field of gastrointestinal physiology. The work refines our understanding of intestinal cell types that contribute to protein uptake and their respective transcriptomes. The work also provides some evidence that microbiomes are modulated by enterocyte protein uptake capacity in a diet-dependent manner. These latter findings provide valuable datasets for future related studies.

We thank the Reviewer for their thorough and kind assessment. We appreciate the suggestion for edits and for pointing out areas that needed further clarification.

One point in need of further explanation is the use *fabp6* (referred to as *fabp2* by the reviewer) to define anterior LREs and their gene expression pattern, which includes high levels of *fabp6*, something that was deemed a “circular argument” by the reviewer. The rationale for using *fabp6* as a reference is that we were able to define its spatial pattern in relation to other LRE markers and the neighboring ileocyte population using transgenic markers (Lickwar et al., 2017; Wen et al., 2021). Thus, far from being a circular argument, using *fabp6* allowed us to identify other markers that are differentially expressed between anterior and posterior LREs, which share a core program that we highlight in our study. In the revised manuscript, we clarified this point (lines 166 – 169).

We followed the Reviewer’s suggestion to test if LRE activity and dietary protein affected beta dispersal. Our analyses revealed that beta dispersion was not significantly different between our experimental conditions. We added details about this analysis (lines 384 – 386) and a new supplemental figure panel (Figure S7C).

**Reviewer #2 (Public review):**
Summary:The authors set out to determine how the microbiome and host genotype impact host protein-based nutrition.Strengths:The quantification of protein uptake dynamics is a major strength of this work and the sensitivity of this assay shows that the microbiome and even mono-associated bacterial strains dampen protein uptake in the host by causing down-regulation of genes involved in this process rather than a change in cell type.The use of fluorescent proteins in combination with transcript clustering in the single cell seq analysis deepens our understanding of the cells that participate in protein uptake along the intestine. In addition to the lysozome-rich enterocytes (LRE), subsets of enteroendocrine cells, acinar, and goblet cells also take up protein. Intriguingly, these non-LRE cells did not show lysosomal-based protein degradation; but importantly analysis of the transcripts upregulated in these cells include dab2 and cubn, genes shown previously as being essential to protein uptake.The derivation of zebrafish mono-associated with single strains of microbes paired with HCR to localize and quantify the expression of host protein absorption genes shows that different bacterial strains suppress these genes to variable extents.The analysis of microbiome composition, when host protein absorption is compromised in cubn-/- larvae or by reducing protein in the food, demonstrates that changes to host uptake can alter the abundance of specific microbial taxa like Aeramonas.Weaknesses:The finding that neurons are positive for protein uptake in the single-cell data set is not adequately discussed. It is curious because the cldn:GFP line used for sorting does not mark neurons and if the neurons are taking up mCherry via trans-synaptic uptake from EECs, those neurons should be mCherry+/GFP-; yet methods indicate GFP+ and GFP+/mCherry+ cells were the ones collected and analyzed.

We thank the Reviewer for the kind and positive assessment of our work, for suggestions to improve the accessibility and clarity of the manuscript, and for pointing out an issue related to a neuronal population that needed further clarification.

It turns out that there is a population of neurons that express *cldn15la*. They are not easily visualized by microscopy because IECs express this gene much more highly. However, the endogenous *cldn15la* transcripts can be found in neurons as shown in a recently published dataset (PMID: 35108531) as well as in this study We added a discussion point to clarify this issue (lines 463 – 465).

**Reviewer #3 (Public review):**
Summary:Childers et al. address a fundamental question about the complex relationship within the gut: the link between nutrient absorption, microbial presence, and intestinal physiology. They focus on the role of lysosome-rich enterocytes (LREs) and the microbiota in protein absorption within the intestinal epithelium. By using germ-free and conventional zebrafishes, they demonstrate that microbial association leads to a reduction in protein uptake by LREs. Through impressive in vivo imaging of gavaged fluorescent proteins, they detail the degradation rate within the LRE region, positioning these cells as key players in the process. Additionally, the authors map protein absorption in the gut using single-cell sequencing analysis, extensively describing LRE subpopulations in terms of clustering and transcriptomic patterns. They further explore the monoassociation of ex-germ-free animals with specific bacterial strains, revealing that the reduction in protein absorption in the LRE region is strain-specific.Strengths:The authors employ state-of-the-art imaging to provide clear evidence of the protein absorption rate phenotype, focusing on a specific intestinal region. This innovative method of fluorescent protein tracing expands the field of in vivo gut physiology.Using both conventional and germ-free animals for single-cell sequencing analysis, they offer valuable epithelial datasets for researchers studying host-microbe interactions. By capitalizing on fluorescently labelled proteins in vivo, they create a new and specific atlas of cells involved in protein absorption, along with a detailed LRE single-cell transcriptomic dataset.Weaknesses:While the authors present tangible hypotheses, the data are primarily correlative, and the statistical methods are inadequate. They examine protein absorption in a specific, normalized intestinal region but do not address confounding factors between germ-free and conventional animals, such as size differences, transit time, and oral gavage, which may impact their in vivo observations. This oversight can lead to bold conclusions, where the data appear valuable but require more nuance.The sections of the study describing the microbiota or attempting functional analysis are elusive, with related data being overinterpreted. The microbiome field has long used 16S sequencing to characterize the microbiota, but its variability due to experimental parameters limits the ability to draw causative conclusions about the link between LRE activity, dietary protein, and microbial composition. Additionally, the complex networks involved in dopamine synthesis and signalling cannot be fully represented by RNA levels alone. The authors' conclusions on this biological phenomenon based on single-cell data need support from functional and in vivo experiments.

We thank the Reviewer for their assessment and for pointing out some areas that needed to be explained better and/or discussed.

The Reviewer mentions some potential confounding factors (ie., size differences, transit time, oral gavage) in the gnotobiology experiments. We would like to convey that these aspects have been addressed in our experimental design and are now clarified in the revised manuscript: 1- larval sizes were recorded and found to be similar between GF and monoassociated larvae (Figure S6A); 2- while intestinal transit time may be affected by microbes and is a topic of interest, in our assay luminal mCherry cargo is present at high levels throughout the gut and is not limiting at any point during the experiment; 3- gavage, which is necessary for quantitative assays, is indeed an experimental manipulation that may somehow alter the subjects (the same is true for microscopy and virtually any research method). However, it cannot explain differences between GF and CV or alter our conclusions via microbial or dietary effects. We now elaborate the former point in the revised discussion (line 426). A new panel has been added for Fig.S6 to show that standard length was similar in GF and monoassociated larvae (Figure S6A).

We are aware that microbial community composition is often highly variable between experiments and this necessitates adequately high biological replication and inclusion of internal controls to allow conclusions to be drawn. Nevertheless, studies evaluating the utility of 16S rRNA gene sequencing have found that this analysis reveals important impacts of environmental factors on the gut microbiome (PMIDs: 21346791, 31409661, 31324413). Our results provide further evidence that 16S rRNA gene sequencing remains a useful method to detect perturbations to the zebrafish gut microbiome. Reproducing previous findings, we detected many of the core zebrafish microbiota strains in our samples that have been identified by other studies (PMIDs: 26339860, 21472014, 17055441). To ensure the robustness of our results, we included several biological replicates for each condition, co-housed genotypes and included large sample sizes to minimize environmental variability between groups. In response to this reviewer concern, we have added a supplemental beta diversity plot and statistical analyses showing that the microbiomes in our larvae were significantly different from the diets or tank water (Figure S7A). This analysis shows that the host environment influenced microbial community composition (lines 376 – 378). We also added an additional supplemental panel and performed analysis showing that the experimental replicates (i.e., different tanks) were not a significant source of variation in this study (lines 378 – 380) (Figure S7B). This result underscores that the microbiota in these larvae were influenced by both the host and diet.

Regarding dopamine pathways, we acknowledge that it involves complex biology that will require dedicated studies. In this work, we simply point out gene expression patterns we find interesting as they may inform future studies.

Finally, the Reviewer mentions the use of inadequate statistical methods for some analyses without specifying or indicating alternative analyses, only the need to justify the use of two-way ANOVA is made explicit. In this point, we respectfully disagree and would like to emphasize that we use statistical methods that are standard in the field (PMID: 37707499). We nevertheless added a justification for the use of two-way ANOVA where appropriate (lines 635-637, 653-654, 773-776). The two-way ANOVA test was to compare fluorescence profiles of gavages cargoes or HCR probes along the length of the LRE region. This test accounts for differences in fluorescence between experimental conditions in segments (30 μm) along the LRE region (~300 μm). This allows us to capture differences in fluorescence between experimental conditions while accounting for heterogeneity in the LRE region. Please see our comment below for more information about our use of the 2-way ANOVA.

**Recommendations for the authors:**

**Reviewer #1 (Recommendations for the authors):**
Please provide in the materials and methods the strain identifiers and sources of the bacteria used in the study.

Thank you for the suggestions. Strain identifiers and source information were added to the methods (lines 576-579).

**Reviewer #2 (Recommendations for the authors):**
(1) This is a very satisfying and thorough analysis of the reciprocal influence of diet, microbiome, and host genotype on protein absorption by the host. Below I make suggestions that mainly relate to making the paper more accessible to a broader audience.(2) Line 233 Starts a section that reports the findings of the scRNA dataset. The writing is inconsistent with respect to how the genes are listed: whether abbreviation only or spelled out followed by abbreviation. I prefer the latter. For example, slc10a2 is a bile acid Na cotransporter but for those not in the know, they would have to look this up. Perhaps adding a supplementary table that provides a gene list of those discussed in the text with abbreviation/spelled-out, and KEGG terms.

Thank you for pointing out inconsistent gene labeling. We have revised the text with spelled out gene names followed by abbreviations.

(3) Line 461 Where did the neurons come from when you were sorting cldn+ cells?

Neuronal expression of *cldn15la* was detected in our data and other published datasets (PMID: 37995681, 35108531). We added a note to the text clarifying that neuronal cells can express *cldn15la* (lines 463-465).

(4) Line 561 1x tricaine should be converted to percentage in solution or concentration throughout.

The tricaine concentration was 0.2 mg/mL. We added this detail to the methods (line 596).

(5) Line 612 Please clarify how normalizations are carried out: is it to the peak value in the germ-free condition? CV never reaches 1.

AUC values were normalized to the peak value in the GF condition at 60 minutes PG. We clarified this step in the methods (lines 618-619).

(6) Line 654-663 I think mCherry here should be mTourquoise?

Thank you for catching this typo. We corrected it in the text.

(7) In Figure 1 Please consider adding a color so that magenta does not represent BOTH germ-free AND mCherry.

Due to the many colors of fluorescent proteins and HCR probes in this paper, we were not able to find an alternative plot line color to represent GF.

(8) In Figure 2 I suggest consistency with respect to the order you present GF/CVFigure 1 GF->CVFigure 2 CV->GFMy preference is GF->CV

Images in Figure 2 were re-ordered following reviewer’s recommendation.

Here, 20 minute time point also appears qualitatively different between GF and CV.

There can be slight differences in LREs between individuals. These images were selected because they represented the average differences in the amount of mTurquoise degradation activity that occurred between 20 – 60 minutes post-flushing in the GF and CV conditions.

In Figure 3E Figure legend refers to being able to see BSA in vacuoles. The image should be modified to show this- currently too small.

In response, we enlarged the confocal microscopy images showing DQ red BSA in the LRE region (Figure 3E). We added a panel with confocal microscopy images of the LREs in 6 dpf larva gavaged with DQ red BSA (Figure S3F). These images show that DQ red BSA fluorescence was localized to the LRE lysosomal vacuole.

In Figure 5D, Posterior LRE should be pink not green in the key to the right of the heatmap.

Thank you for catching this error. We have corrected the colors (Figure 5D).

**Reviewer #3 (Recommendations for the authors):**
(1) Introduction and context:

Expand the introduction to include more background on microbial-mediated protein absorption, with references to relevant findings in *Drosophila*. This will provide a stronger foundation for the study's contributions to the field.

Thank you for this suggestion. We added information about microbe-mediated amino acid harvest in *Drosophila* to the introduction (lines 49-53).

(12) Methodological suggestions:Measure and report differences between germ-free (GF) and conventional (CV) animals, such as transit time, to account for potential confounding factors in protein absorption dynamics.

We respectfully assert that a transit assay is not required for this study and could actually create confusion as an effect in transit time could be interpreted as a contributing factor when it is in fact not the case due to the experimental design. This is because the concentration of luminal protein was equivalent in GF and CV larvae (Figure S1E), so the LREs had equal saturating access to those proteins in both conditions. Furthermore, we showed the microbiota did not degrade fluorescent protein (Figure S1F). Therefore, we feel confident that there was lower protein uptake in the LREs of CV larvae because the microbiome exerted regulatory effects on LRE activity.

Provide detailed information on the gating strategy used for single-cell sorting to enhance the dataset's utility and support claims about cell changes.

The methods we used for sorting cells were previously described (PMID: 31474562). In this manuscript, we describe them under the heading “Fluorescence activated cell sorting for single cell RNA-sequencing.”

Explain the "GeneRatio" metric in figure legends for clarity.

The GeneRatio is the ratio of genes associated with each individual GO term to the number of genes associated with the domain. An explanation was added to the caption (Figure S3C).

(13) Visual and statistical improvements:Include images of labeled peptidases within lysosome-rich enterocytes (LREs) to reinforce findings.

Thank you for the suggestion. We added images of labeled peptidases in the LRE region (Figure S6E-D).

For Panels 4-F and 5-D, consider using violin plots of selected genes to improve clarity and emphasize major ideas.

In Figure 4F, the heatmap shows multiple genes were upregulated in mCherry-positive cells. We tried the plotting suggested by the reviewer and felt that violin plots could not convey this message as clearly. Likewise, the heatmap in Figure 5D effectively shows the gradient of expression between ileocytes, anterior and posterior LREs.

Strengthen statistical analysis by employing more rigorous methods and justifying their selection, such as using two-way ANOVA where appropriate.

The two-way ANOVA was used to quantify protein uptake or HCR probe fluorescence along the length of the LRE region. This statistical test allowed us to compare differences in fluorescence between experimental conditions in multiple LRE segments (see Authoer response image 1 below for example). As our assays show, the LRE region is heterogenous with segments showing different levels of activity and gene expression. The two-way ANOVA is appropriate because it allows us to account for this heterogeneity by comparing fluorescence across multiple segments.

**Author response image 1. sa4fig1:** 

Our figures display these fluorescent levels in line plots (above, left) rather than bar plots (above, right). The results are easier to visualize interpret in line plots, and they display the fluorescence profiles in greater detail.

(14) Technical corrections:Correct figure references: Figure 5 about tryptophan metabolism should be 5A, S5G-S5H.

We corrected the figure references.

Line 518: Spell out "heterozygotes" instead of using "gets".

We changed the term from “hets” to “heterozygotes.”

(15) Revise Figure S2 citation to match the actual figure labeling.

We corrected the text to indicate “Figure S2” rather than “Figure S2A.”

Additional manuscript modification

· Figure panels 3B-C, S3A-B, 4A-C: Two cluster were relabeled with improved descriptors based on our updated annotations. The clusters “Pharynx-esophagus-cloaca 1” (PEC1) and PEC2 were relabeled as “Pharynx-cloaca 1” and “Pharynx-cloaca 2.”